Reassessment of pore occlusion in some diatom taxa with re-evaluation of Placoneis Mereschkowsky (Bacillariophyceae: Cymbellales) and description of two new genera

http://orcid.org/0000-0001-9936-0652 Mironov Andrei 1 2 3 diatomironov@yandex.ru
Glushchenko Anton 2
http://orcid.org/0000-0003-4710-319X Maltsev Yevhen 2
Genkal Sergey 4
Kuznetsova Irina 2
http://orcid.org/0000-0001-9824-7164 Kociolek John Patrick 5
Liu Yan 1
Kulikovskiy Maxim 2
1 College of Life Science and Technology, Harbin Normal University , Harbin, Heilongjiang Province , China
2 Laboratory of Molecular Systematics of Aquatic Plants, K. A. Timiryazev Institute of Plant Physiology RAS , Moscow , Russia
3 Department of Mycology and Algology, M. V. Lomonosov Moscow State University , Moscow , Russia
4 Papanin Institute for Biology of Inland Waters RAS , Borok, Yaroslavl Region , Russia
5 Museum of Natural History , Boulder, Colorado , United States
Gillespie Joseph
Electronic publication date: 2024 May 29
Publication date: 2024
Volume: 12
Electronic Location ID: e17278
Received 2024 Jan 2; Accepted 2024 Apr 1
Copyright: © 2024 Mironov et al.
Copyright year: 2024
Copyright holder: Mironov et al.
License: This is an open access article distributed under the terms of the Creative Commons Attribution License, which permits unrestricted use, distribution, reproduction and adaptation in any medium and for any purpose provided that it is properly attributed. For attribution, the original author(s), title, publication source (PeerJ) and either DOI or URL of the article must be cited.
License URL: https://creativecommons.org/licenses/by/4.0/

Keywords: Diatoms, Cymbellales, Placoneis, Tectulum, Pore occlusion, Morphology, Taxonomy, New genera, New species

Funding: Russian Science Foundation 23-74-10081 National Natural Science Foundation of China 31970213 Ministry of Science and Higher Education of the Russian Federation 122042700045-3 This publication is based on research carried out with financial support from the Russian Science Foundation (23-74-10081) for sample investigation, morphological distributions and molecular research and the National Natural Science Foundation of China (31970213) for LM and SEM and by the framework of state assignment of the Ministry of Science and Higher Education of the Russian Federation (theme 122042700045-3) for finishing the manuscript. The funders had a role in data collection and analysis, and preparation of the manuscript. The funders had no role in the study design and decision to publish.

==============================
In this article, the history and taxonomy of Placoneis gastrum, the type species of the genus Placoneis, was discussed. We investigated the structure of pore occlusions in Placoneis and related genera. As a result, we propose a new classification for tectulum-like types of pore occlusions. The new classification is congruent with previously-published and newly-constructed phylogenies based on molecular data. Based on the different structures of the pore occlusions, species of Placoneis are transferred to Witkowskia gen. nov. Hence, 168 new combinations are introduced. A new diatom species, with a similar morphology to Placoneis flabellata, was discovered in Bắc Kạn Province, Vietnam. It is described in this article as Chudaevia densistriata sp. nov. Placoneis flabellata is transferred to Chudaevia gen. nov. We also illustrate Placoneis flabellata herein and compare it to Chudaevia densistriata sp. nov. An unknown diatom, similar to Placoneis coloradensis, was discovered in Chukotka, Russia. It is introduced as Placoneis elinae sp. nov. below. Additionally, we discuss the distribution of some species of Witkowskia gen. nov. and Chudaevia gen. nov.

Introduction

Placoneis Mereschkowsky is a species-rich genus of diatoms with repeatedly revised taxonomy during the last century. It was erected in 1903 to include numerous species from Navicula Bory sensu lato with a single asymmetrical chloroplast. Mereschkowsky (1903) also studied the structure of plastids and pyrenoids among cymbelloid diatoms. Based on chloroplast morphology, he suggested that Placoneis should be included in the group Monoplacatae alongside cymbelloid and gomphonemoid diatoms. However, since its original description in 1903, Placoneis was consistently wrongly placed in Navicula sensu lato due to its naviculoid symmetry. It was not until 1987, when E.J. Cox highlighted this situation and described morphological similarities between species of Placoneis and cymbelloid diatoms (Cox, 1987). Besides, Round, Crawford & Mann (1990) placed Placoneis in the family Cymbellaceae that also included genera Cymbella Agardh, Brebissonia Grunow, Encyonema Krammer and Gomphocymbella O. Müller. Round, Crawford & Mann (1990) postulated that according to the frustule structure and protoplast characters Placoneis is closely related to Cymbella and Gomphonema Ehrenberg. Close relationships between Placoneis and members of Cymbellales D.G. Mann in Round, Crawford & Mann (1990) were later supported on the basis of morphological, phylogenetic and molecular analyses (Bruder & Medlin, 2007; Kermarrec et al., 2011, Kulikovskiy et al., 2014a; Kezlya et al., 2020, 2021, 2022).

The generitype of Placoneis – P. gastrum (Ehrenberg) Mereschkowsky–was chosen by Cox (1987). Later, Cox (2003) studied the original material from Ehrenberg in search of P. gastrum but was unable to locate specimens. However, she identified some valves of Navicula amphibola P.T. Cleve and indicated that this species is morphologically similar to P. gastrum. Thus, she lectotypified P. gastrum from Donkin’s (1873) material “since its modern usage can be traced back unequivocally to Donkin (1873)”. On the contrary, Jahn (2004) found P. gastrum in Ehrenberg’s type material and proved that the previous lectotypification was illegal. Thus, in the modern conception P. amphibola (P.T. Cleve) Cox, designated from Navicula amphibola by Cox (2003), is treated as a junior heterotypic synonym of P. gastrum (Reichardt, 2018).

It is worth mentioning that Cox (1987) described internal pore occlusions in Placoneis as volae. This term is treated differently by various taxonomists. For instance, Ross & Sims (1972) described vola as “a part of velum that consists of a number of elements projecting from the wall of the poroids or loculus but not fusing”. Mann (1984), on the other side, understood vola as “a flap of silica, attached to the wall of the pore by a fairly broad base and extending most of the way across it, leaving only a curved slit”. It is not clear, what definition of vola was used by Cox when she revaluated Placoneis (Cox, 1987), because the structure of pore occlusions in new generitype Placoneis gastrum was not illustrated with SEM microphotographs. In 2004 she tried to solve this confusion by proposing an idea to understand vola sensu (Ross & Sims, 1972; Cox, 2004). Besides, two new types of pore occlusions were introduced in addition to three widely used velum types (cribrum, rota and vola): foricula (to replace vola sensu D.G. Mann) and tectulum. The latter term was established primarily to describe pore occlusions of Placoneis. So, tectulum is “an inner round or squarish flap-like covering, attached to the edges of an areola by several, regularly arranged, small struts” (Cox, 2004).

A new type of pore occlusion was discovered in an unknown species of Placoneis from Chukotka, Russia (described here as Placoneis elinae Kulikovskiy, Mironov, Genkal, Glushchenko & Kociolek sp. nov.). This type of occlusion is different from the tectulum and is described below as pseudotectulum. The same type of occlusion can be found in three species with similar morphology–Placoneis amphibola, Placoneis coloradensis Kociolek & E.W. Thomas and Naviculadicta amphiboliformis Metzeltin, Lange-Bertalot & Nergui. Thus, Witkowskia Kulikovskiy, Mironov, Glushchenko & Kociolek gen. nov. is introduced in this article to include all species of Placoneis that can be characterized by the presence of tectulum.

In addition, we studied a species from South-East Asia, Navicula flabellata F. Meister, and another morphologically similar species, which is new to science. Both species possess a new type of pore occlusion, named below as paratectulum. To include the species with this unique type of pore occlusion, we describe Chudaevia Kulikovskiy, Mironov, Glushchenko & Kociolek gen. nov. with Chudaevia densistriata Kulikovskiy, Mironov, Genkal, Glushchenko & Kociolek sp. nov. and propose a new combination Chudaevia flabellata (F. Meister) Kulikovskiy, Glushchenko, Mironov & Kociolek comb. nov. In addition, we discuss the variety of pore occlusions among genera of the order Cymbellales.

Materials and Methods

List of samples and corresponding slides is given in Table 1. The samples were acquired and then treated according to the method, described in Glushchenko et al. (2023). Specifically, the samples were treated with 10% hydrochloric acid to remove carbonates and washed several times with deionized water every 12 h. Afterwards, samples were boiled in concentrated hydrogen peroxide (~37%) to mineralize the organic matter. They were washed again with deionized water four times at 12-h intervals. After decantation and filling with deionized water up to 100 ml, the suspension has been spread onto cover slips and left to dry at room temperature.

Table 1 List of the collected samples.

Sample no.	Slide no.	Locality	Coordinates	Substratum	Date of collection	
	Laos	
8	00955	Vientiane Province, Van Vieng Dustrict, Nam Lik River	18°36.808′ N; 102°24.605′ E	Periphyton	24 November 2011	
49	00996	Khammouane Province, unnamed spring	18°12.512′ N; 104°31.507′ E	Periphyton	28 November 2011	
	Vietnam	
BB 1/10	02168	Bắc Kạn Province, Ba Bể Lake	22°33.083′ N; 105°50.267′ E	Benthos	29 April 2015	
	Russia	
B445	13899	NE Siberia, Chukotka Peninsula, small unnamed lake below Markovo Township	64°41.039′ N; 170°24.116′ E	Benthos	18 August 1980	
Note:

Information about the samples used in the current research.

Permanent diatom preparations were mounted in Naphrax® (Brunel Microscopes Ltd., Chippenham, UK; refractive index = 1.73). Light microscopic (LM) observations were performed with a Nikon Eclipse E600 equipped with Plan-apochromatic oil immersion objective x100 (n.a. 1.4) and a Nikon DS-5M digital camera, as well as a Zeiss Axio Scope A1 (Carl Zeiss Microscopy GmbH, Göttingen, Germany) microscope equipped with an oil immersion objective (Plan-apochromatic ×100/n.a.1.4, Nomarski differential interference contrast, DIC) and a Zeiss Axio Cam ERc 5s camera (Carl Zeiss NTS Ltd., Oberkochen, Germany). Valve ultrastructure was examined by means of a Hitachi S-4500 field emission scanning electron microscope (Hitachi Co., Ltd., Tokyo, Japan) and JSM-6510LV scanning electron microscope (JEOL Ltd., Tokyo, Japan) operated at 10 kV and 11 mm working distance. For scanning electron microscopy (SEM), parts of the suspensions were fixed on aluminum stubs after air drying. The stubs were sputter-coated with 50 nm of gold in an Eiko IB 3 (Eiko Engineering Co., Ltd., Hitachinaka, Japan).

Molecular analysis, performed herein, was carried according to the algorithm, performed in Glushchenko et al. (2023). Thus, the dataset consisted of concatenated 82 SSU rDNA, and 85 rbcL sequences, selected for available Cymbellales lineages and five diatom species from Rhopalodiaceae chosen as the outgroups (taxa names and accession numbers are given in Fig. 1). The SSU rDNA and rbcL sequences were aligned separately using the G-INS-I algorithm in the Mafft ver. 7 software (RIMD, Osaka, Japan) (Katoh & Toh, 2010). The resulting data set comprised of 1,731, and 1,401 nucleotide sites for nuclear SSU rDNA, and plastid rbcL regions, respectively. After removal of the unpaired regions, the aligned SSU rRNA gene sequences were combined with the rbcL gene sequences into a single matrix for concatenated SSU rDNA and rbcL.

Figure 1 Phylogenetic position of Witkowskia gen. nov., Paraplaconeis and Geissleria species based on Bayesian inference from an alignment of 90 sequences and 3,132 characters (rbcL. SSU rRNA genes).

Values of likelihood bootstrap (LB) from ML analyses below 50 are not shown. Values of Bayesian posterior probabilities (PP) below 0.9 are not shown. Strain numbers (if available) and GenBank numbers are indicated for all sequences.

The Bayesian inference (BI) method was performed using Beast ver. 1.10.1 software (BEAST Developers, Auckland, New Zealand) (Drummond & Rambaut, 2007). The most appropriate partition-specific substitution models, shape parameter α and a proportion of invariable sites (pinvar) were recognized by the Bayesian information criterion (BIC) in jModelTest ver. 2.1.10 software (Vigo, Spain) (Darriba et al., 2012). This BIC-based model selection pro-cedure selected the following models, shape parameter α and a proportion of invariable sites (pinvar): GTR + G + I, α = 0.4710 and pinvar = 0.5970 for SSU rDNA; TPM1uf + G + I, α = 0.3960, and pinvar = 0.7310 for the first codon position of the rbcL gene; JC + I, pinvar = 0.8690 for the second codon position of the rbcL gene; GTR + G + I, α = 1.1260, and pinvar = 0.2320 for the third codon position of the rbcL gene. However, the HKY model was applied instead of TPM1uf, and the F81 applied instead of JC as the most similar applicable options for BI. A speciation model was performed by a Yule process tree prior. Five MCMC analyses were run for 5 million generations (burn-in 1,000 million generations). The convergence diagnostics was performed in the Tracer ver. 1.7.1 software (MCMC Trace Analysis Tool, Edinburgh, UK) (Drummond & Rambaut, 2007). The initial 15% trees were removed, the rest retained to construct a final chronogram with 90% posterior probabilities. The robustness of tree topologies was assessed by boot-strapping the data set with Maximum Likelihood (ML) analysis using RaxML software (Stamatakis, Hoover & Rougemont, 2008). The ML bootstrapping was performed with 1,000 replicas. Trees were viewed and edited using FigTree ver. 1.4.4 (University of Edinburgh, Edinburgh, United Kingdom) and Adobe Photoshop CC ver. 19.0 software (San Jose, CA, USA).

Results

Phylogeny of the Cymbellales is presented in Fig. 1. The tree shows this order to be monophyletic. The genera Cymbella C.A. Agardh, Didymosphenia A. Schmidt, Karthickia Kociolek, Glushchenko & Kulikovskiy, Encyonopsis Krammer, Encyonema Kützing and Cymbopleura (Krammer) Krammer are represented in lineages that together represent a grade of organization rather than a monophyletic clade. The clade of Encyonema species is the sister taxon to a clade comprised of Geissleria Lange-Bertalot & Metzeltin, Paraplaconeis Kulikovskiy, Lange-Bertalot & Metzeltin, & Witkowskia gen. nov., which together is sister to a clade of gomphonemoid diatoms, including Gomphonella Rabenhorst, Reimeria Kociolek and Stoermer, Gomphonema Ehrenberg and Gomphoneis P.T. Cleve. In the clade containing Geissleria, Paraplaconeis & Witkowskia gen. nov., which is strongly supported (Bayesian Post Priori = 0.98), each of these genera are also shown to be monophyletic with support ranging from strong to low for each.

Placoneis Mereschkowsky

Type species: Placoneis gastrum (Ehrenberg) Mereschkowsky

Synonyms: Navicula amphibola Cleve, 1891; Placoneis amphibola (Cleve) Cox, 2003.

Description. Cells solitary, rectangular in girdle view, with a single H-shaped chloroplast. It consists of two elongated arms, connected by a narrow isthmus in the center of the valve. Valve outline variable, from linear-elliptic to lanceolate, apices acutely or broadly rounded, subcapitate, subrostrate, capitate or rostrate. Raphe often filiform, proximal raphe ends expanded and curved to the same or opposite sides. Distal raphe ends curved to the same side. In some species it seems that they are oppositely curved, but in SEM it is clear that one of the terminal fissures recurves and forms a hook-like ending. Intermissio present. Axial area usually narrow, central area variable. Isolated pores (stigmoids) may be present (up to four on the same valve). Their openings are round externally and round or oval internally. Striae uniseriate, sometimes shortened and more widely spaced in the central valve region. Areolae round externally and usually square internally. Areolae are occluded by pseudotectula.

Placoneis elinae Kulikovskiy, Mironov, Genkal, Glushchenko & Kociolek sp. nov. (Figs. 2–5)

Figure 2 (A–D) Placoneis elinae sp. nov. (Fresh) Sample no. B445 (corresponds material in the slide no. 13899).

Light microscopy. Live cells with chloroplast structure. Size diminution series. Scale bar = 10 µm.

Figure 3 (A–H) Placoneis elinae sp. nov. Slide no. 13899.

Light microscopy, size diminution series. (D) Holotype. Scale bar = 10 µm.

Figure 4 (A–E) Placoneis elinae sp. nov. Sample 13899 (corresponds material in the slide no. 13899).

Scanning electron microscopy, external view. (A, B) Whole valve. (C) Central area. White arrows show the proximal raphe ends. Black arrows show the stigmoids. White arrowhead shows the areolar depression. (D) Valve mantle. White arrowhead shows less developed areolar depression. (E) Valve end. White arrow shows the distal raphe end. Black arrow shows the areolae extending to valve margin. White arrowhead shows less developed areolar depression. Scale bars (A, B) 10 µm (C) 3 µm (E) 3.5 µm (D) 1.5 µm.

Figure 5 (A–E) Placoneis elinae sp. nov. Sample 13899 (corresponds material in the slide no. 13899).

Scanning electron microscopy, internal view. (A) Whole valve. (B) Central area. White arrows show the internal slits of stigmoids. Black arrows show the proximal raphe ends. (C) Valve end. Black arrow shows the distal raphe end. (D, E) Areolae, occluded by pseudotectula. Black arrows show occlusions of areolae. Scale bars (A) 10 µm (B) 3 µm (C) 2.5 µm (D, E) 1 µm.

Holotype here designated: Slide no. 13899, Fig. 3D, deposited in herbarium of MHA, Main Botanical Garden, Russian Academy of Science, Moscow, Russia.

Isotype. Slide no. 13899 m, collection of Maxim Kulikovskiy at the Herbarium of the Institute of Plant Physiology, Russian Academy of Science, Moscow, Russia.

Type locality. Russia, NE Siberia, Chukotka Peninsula, small unnamed lake below Markovo Township (64°41.039′ N; 170°24.116′ E), leg. S.I. Genkal, 18 August 1980.

Etymology. The species name is dedicated to the wife of Andrei Mironov–Elina.

Distribution. The species is yet known only from its type locality.

Description. LM (Figs. 2A–2D, 3A–3E). Cells solitary. Chloroplast has a typical organization inherent in representatives of the genus, being a single H-shaped plastid, with one arm lying against each side of the girdle, connected by a narrow central isthmus (Figs. 2A–2D). Valves linear-elliptic to broadly linear-lanceolate. Valve ends rostrate to subcapitate. Length 44.5–74.0 µm, breadth 21.0–26.5 µm. The axial area is narrow at the poles, gradually becoming wider towards the center. Central area bowtie-shaped, formed by 3–5 shortened striae on each side. Raphe undulate, lateral, becoming filiform towards the central nodule. Proximal raphe ends expanded and rounded. Distal raphe ends extend to valve margin. Usually, four stigmoids are present in the central area. Striae distinctly radiate throughout the valve, 8.5–10.0 in 10 µm in the central portion, up to 11.0 in 10 µm near the apices. Areolae 14.0–21.0 in 10 µm, well-resolvable in LM, appear to be rectangular and slightly apically elongated (Figs. 3A–3H).

SEM, external view (Figs. 4A–4E). Valve face is flat. Raphe undulate, raphe slit expanding near the proximal ends. Proximal raphe ends oval, slightly offset (Fig. 4C, white arrows). Distal raphe ends hook onto the valve and terminate on valve mantle (Fig. 4E, white arrow). Four stigmoids are distinguishable at the end of striae, adjacent to the central area (Fig. 4C, black arrows). Striae uniseriate, composed of rounded areolae, extending to valve margin (Fig. 4E, black arrow). Unlike stigmoids, areolae in striae lie in regular, shallow, hourglass-shaped depressions (Fig. 4C, white arrowhead). These depressions are less developed near the apices and on the valve margin (Figs. 4D, 4E, white arrowheads). Due to depressions, areolae seem to be apically elongated in LM.

SEM, internal view (Figs. 5A–5E). The raphe is straight. Proximal raphe ends small, hook-shaped, deflected to one side (Fig. 5B, black arrows). Distal ends terminating on small helictoglossae, slightly deflected to the opposite side from proximal raphe ends (Fig. 5C, black arrow). Internal slits of stigmoids are relatively broad, aligned with striae (Fig. 5B, white arrow). Stigmoids are not separated from the rest of stria. Rows of areolae are narrower than virgae, vimines very slender. Areolae rectangular or rectangular-elliptical, occluded by pseudotectula (Figs. 5D, 5E, black arrows). Each pseudotectulum is formed by several elongated struts of irregular shape and orientation. It is very likely that struts are equipped with flap-like coverings, that demolished during the sample preparation.

Differential diagnosis. Morphologically, the new species is most similar to Placoneis coloradensis. However, P. elinae has more linear valve margins and more capitate apices. Also, the new species can be distinguished by a bowtie-shaped central area and shallow depressions of areolae (see Table 2). Placoneis amphiboliformis (Metzeltin, Lange-Bertalot and Soninkhishig) Vishnyakov (syn. Naviculadicta amphiboliformis) differs from P. elinae by less-protracted apices and elongated areolar openings without shallow depressions (see Table 2).

Table 2 Comparison of morphological features of Placoneis elinae sp. nov., Placoneis gastrum and related species.

	Pl. elinae sp. nov.	Pl. coloradensis	Naviculadicta amphiboliformis	Pl. gastrum s.s. lectotypus	Pl. amphibola	Pl. amphibola	Pl. gastrum	Pl. gastrum	Pl. gastrum	Pl. gastrum	
Note	Our material	Kociolek and Thomas data	Metzeltin et al. data	Reichardt data	sensu Cox	sensu Metzeltin et al.	sensu Jahn	sensu Kulikovskiy et al.	sensu Chudaev and Gololobova	sensu Lange-Bertalot et al.	
Valve outline	Linear-elliptic to broadly linear-lanceolate	Linear-elliptic to broadly linear-lanceolate*	Elliptical	Elliptical to broadly-lanceolate*	Broadly lanceolate	Elliptical to broadly-lanceolate*	Elliptical, broadly lanceolate*	Elliptical, weakly dorsiventral	Elliptical, weakly dorsiventral*	Broadly lanceolate, slightly asymmetrical	
Valve ends	Rostrate to subcapitate	Rostrate to subcapitate*	Rostrate to subrostrate	Broadly rounded, subrosrtate*	Broadly rounded, subrosrtate*	Broadly rounded, subrosrtate *	Rostrate to subrosrtate*	Rostrate to broadly rounded	Rostrate to broadly rounded*	Broadly protracted, obtusely rounded	
Axial area	Narrow, slightly expanded to the central part of valve	Narrow, expanded to the central part of valve	Medium sized, slightly expanded to the central part of valve*	Narrow, slightly expanded to the central part of valve*	Narrow, not expanded*	Narrow, not expanded*	Narrow, not expanded*	Narrow, slightly arched	Narrow, slightly arched*	Narrow, slightly expanded to the central part of valve*	
Central area	Bowtie-shaped, with 4 stigmoids	Rectangular, with several isolated areolae	Bowtie-shaped, stigmoids unresolvable*	Rectangular, with 4 stigmoids*	Rounded to diamond-shaped	Rectangular, with 4 stigmoids*	Transversely widened*	Roundish to rectangular	Small, roundish*	Transversely widened	
Valve length, μm	44.5–74.0	65–80	34–63	40–56	30–60	34–63	42*	30–58	40.4–56.2	30–60	
Valve breadth. μm	21.0–26.5	29–31	18–24	20–24	12–18	22–29	19*	15–20	17.3–19.3	12–18	
Raphe	Undulate, filiform, lateral towards the ends	Undulate, lateral	Undulate, lateral*	Undulate, lateral*	Undulate, lateral, slightly curved*	Undulate, lateral*	Narrow, filiform*	Narrow, filiform, weakly arched	Narrow, filiform, weakly arched*	Narrow, filiform	
Striae	Radiate throughout. Present irregularly shortened striae on each side	Radiate throughout, parallel at the apices. Present 5–7 shortened striae on each side	Radiate throughout	Radiate throughout, parallel at the apices. Present 3–4 shortened striae on each side*	Radiate throughout. Present several shortened striae on each side	Radiate throughout, slightly curved. Present 3–4 shortened striae on each side*	Radiate throughout. Present several irregularly shortened striae on each side*	Radiate throughout. Present several irregularly shortened striae on each side	Radiate throughout. Present 3–4 irregularly shortened striae on each side	Radiate throughout, narrow. Present irregularly shortened striae on each side	
Striae in 10 μm	8.5–10.0	5–6	(7) 8–10	7–9	8–10	n.d.	8.5–9.0	6–10	(6.8) 6.9–7.0 (8.4)	8–10	
Areolae in 10 μm	14–21	n.d.	10–12	16	n.d.	n.d.	n.d.	n.d.	23.4–24.8	22–24	
Distribution	Recent. Russia, Chukotka	Recent. USA, Colorado.	Recent. Central Asia. Mongolia, Baruun burkh River	Recent. Central Europe. Germany	Fossil and recent. Widely distributed.	Recent. Northern Europe. Norway, Spitzbergen, Bear Island	Recent and fossil. Mexico, Vera-Cruz. USA, Connecticut. Iceland, Husavic	Recent. Widely distributed	n.d.	Recent. Central Europe. Germany	
References	This study	Kociolek & Thomas (2010)	Metzeltin, Lange-Bertalot & Soninkhishig (2009)	Reichardt (2018)	Cox (2003)	Metzeltin, Lange-Bertalot & Soninkhishig (2009)	Jahn (2004)	Chudaev & Gololobova (2016)	Kulikovskiy et al. (2016)	Lange-Bertalot et al. (2017)	
Note:

New species are compared with related taxa by morphology and distribution.

* Counted from published data; n.d.—no data.

Witkowskia Kulikovskiy, Mironov, Glushchenko & Kociolek gen. nov.

Type species: Witkowskia neohambergii (Glushchenko, Kezlya, Kulikovskiy & Kociolek) Kulikovskiy, Glushchenko, Mironov & Kociolek comb. nov.

Description. Cells symmetrical, with variously rostrate to capitate apices. Girdle relatively shallow. Each cell has a single chloroplast with a central pyrenoid. The plastid is formed by two X-shaped plates connected by an isthmus. Central area usually expanded. Isolated pores (stigmoids) may be present near the central area. Axial area narrow. Distal raphe endings curved to the same side of the valve (sometimes this feature is discernable only in SEM). In SEM, proximal endings appear to be straight externally and laterally deflected internally. Striae uniseriate, often slightly radiate around the valve center, more or less parallel towards the apices, composed of small rounded poroids. Areolar openings round to elongated externally, internally occluded by tectula–round or squarish flap-like coverings, equipped with several small struts. Struts are regularly arranged and directed perpendicular to the surface of the valve or may be slightly tilted relative to the areolar opening (Kezlya et al., 2021).

Etymology. The genus is named in honor of Professor Dr Andrzej Witkowski (1955–2023), a prominent diatomist from Poland, for his contributions to understanding of diatom systematics, morphology and evolution.

Chudaevia Kulikovskiy, Mironov, Glushchenko & Kociolek gen. nov.

Type species: Chudaevia densistriata Kulikovskiy, Mironov, Genkal, Glushchenko & Kociolek sp. nov.

Description. The structure of the chloroplast is unknown. Cells symmetrical, linear-elliptic to elliptic. Valve apices rounded (in the post-initial valves) to rostrate. Axial area narrow, slightly expanded in the central part of the valve. Central area small, rounded to rhomboid, formed by shortened striae on each side. Raphe filiform to weakly. Proximal raphe ends drop-shaped, distal raphe ends extend to valve margin. A single isolated stigmoid located near the central raphe ends, slightly offset laterally relative to the axial area. Striae uniseriate, almost parallel at the central part of valve, then radiate throughout. Striae narrower than virgae. Areolae externally apically elongated, vimines with warty outgrowths. Areolae internally occluded by paratectula—flap-like coverings, equipped with 3–4 small struts. Struts regular, directed parallel to the valve surface.

Etymology. The genus is dedicated to our colleague, diatomist from Moscow State University, Dr Dmitry Alekseevich Chudaev.

Chudaevia densistriata Kulikovskiy, Mironov, Genkal, Glushchenko & Kociolek sp. nov. (Figs. 6–8)

Figure 6 (A–K) Chudaevia densistriata sp. nov. Slide no. 02168.

Light microscopy, size diminution series. (I) Holotype. Scale bar = 10 µm.

Figure 7 (A–E) Chudaevia densistriata sp. nov. Sample 02168 (corresponds material in the slide no. 02168).

Scanning electron microscopy, external view. (A, B) Whole valve. Black arrows show the proximal raphe ends. (C) Cental area with a stigmoid. White arrow shows the rounded areolae. Black arrow shows the isolated stigmoid. (D, E) Valve ends. White arrow shows the elongated areolae. Black arrows show the heteromorphic distal raphe ends. Scale bars (A) 10 µm (B) 5 µm (C, D) 1 µm (E) 0.5 µm.

Figure 8 Chudaevia densistriata sp. nov. Sample 02168 (corresponds material in the slide no. 02168).

Scanning electron microscopy, internal view. (A, B) Whole valve. White arrow shows the slit of stigmoid. Black arrow shows the raphe on a raised sternum. (C) Central area. White arrow shows the slit of stigmoid. Black arrows show the proximal raphe ends. (D) Valve end. White arrow shows the distal raphe end terminating with a helictoglossae. Black arrow shows areolar occlusions. (E) Note areolae with paratectula and vimines with warty outgrowths. Black arrow shows areolar occlusion. Scale bars (A, B) 5 µm; (B) 5 µm; (C, D) 1 µm; (E) 0.5 µm.

Holotype here designated: Slide no. 02168, Fig. 6I, deposited in herbarium of MHA, Main Botanical Garden, Russian Academy of Science, Moscow, Russia.

Isotype. Slide no. 02168 m, collection of Maxim Kulikovskiy at the Herbarium of the Institute of Plant Physiology, Russian Academy of Science, Moscow, Russia.

Type locality. Vietnam, Bắc Kạn Province, Ba Bể Lake (22°33.083′N; 105°50.267′ E), leg. M.S. Kulikovskiy, 29 April 2015.

Etymology. The specific epithet refers to the dense striae arrangement.

Distribution. South-East Asia: Vietnam (Lake Ba Bể, type locality). East Asia: South Korea: Nakdong River estuary (as Placoneis flabellata) (Joh, 2013); Japan, Biwa Lake (as Navicula diversipunctata Hustedt) (Tuji, 2003).

Description. LM (Figs. 6A–6K). The structure of the chloroplast is unknown. Valves linear-elliptic to elliptic. Valve ends rounded (in the post-initial valves) to rostrate. Length 25.5–64.0 µm, breadth 15.0–19.5 µm. Axial area narrow, slightly expanded to the central part of valve. Central area rounded to rhomboid. In the center of valves are present irregularly shortened striae on each side. Raphe narrow, filiform, weakly lateral towards the ends. Proximal raphe ends drop-shaped. Distal raphe ends extend to valve margin. One stigmoid located closely to the central raphe ends, slightly offset laterally relative to the axial area; at the post-initial valves is weakly expressed. Striae almost parallel at the central part of valve, then radiate throughout, slightly undulated, 13–15 in 10 µm. Areolae not resolvable in LM (Figs. 6A–6K).

SEM, external view (Figs. 7A–7E). Valve face is flat. Raphe weakly lateral towards the ends. Proximal raphe ends small, with loop-like depressions that are slightly offset from each other (Fig. 7B, black arrows). Distal raphe ends are heteromorphic: one is simple, the second–is hooked (Figs. 7D, 7E, black arrows). Distal raphe ends terminate on valve mantle. Stigmoid isolated, round (Fig. 7C, black arrow). Striae uniseriate, composed of rounded (at the central area, Fig. 7C, white arrow) or elongated areolae (at the ends, Fig. 7D, white arrow), extending to valve margin. Areolae 48–50 in 10 µm.

SEM, internal view (Figs. 8A–8E). The raphe is straight, located on a raised sternum (Fig. 8A, black arrow). Proximal raphe ends are hook-shaped and deflected to one side, towards the stigmoid (Fig. 8C, black arrows). Distal ends terminating on small helictoglossae and slightly deflected to opposite sides from proximal raphe ends (Fig. 8D, white arrow). Stigmoid opening slit-like, located on raised central nodule (Figs. 8A, 8C, white arrow). Slit opening of the stigmoid is also noticeable in LM with careful focusing. Areolae arranged in apically elongated series compared to the wide interstriae (virgae). Warty outgrowths are present between the areolae (the vimines bear small silica ridges interrupted in the middle). Areolae elliptical (rarely round) with 2–4 projections (struts) extending into the lumen of the openings, thus forming a paratectulum. The projections create an illusion of S-shaped opening of areolae (Figs. 8D, 8E, black arrow). Areolar occlusions are similar to tectula but composed of struts that lie parallel to the valve surface (not perpendicular).

Differential diagnosis. Morphologically, the new species is most similar to Placoneis flabellata. However, Chudaevia densistriata can be distinguished by the greater density of striae and more regular rhomboid central area (see Table 3). A species, detected and illustrated as Navicula flabellata by Moisseeva (1971), is morphologically similar to Chudaevia densistriata sp. nov. by valve outline, slightly undulated striae and the presence of stigmoid (see Table 3). Two finds of this species of this species beyond its type locality is similar to the material, described below (Table 3). Moisseeva (1971) stated that in her material specimens of N. flabellata differ from Hustedt’s illustrations (1937a) by more delicate striae (Moisseeva, 1971: p. 76). The compared species also differ in valve width (15.0–19.5 μm in Chudaevia densistriata sp. nov. against 13–15 μm in Navicula flabellata sensu Moisseeva). Striae density is also different (13–15 in 10 μm in Chudaevia densistriata sp. nov. against 15–16 in 10 μm in Navicula flabellata sensu Moisseeva). The species, illustrated by Tsoy (2017) and indentified as Placoneis flabellata, is also similar to the newly described species by valve outline, striae arrangement and the presence of a single stigmoid. (Table 3) (Tsoy, 2017: p. 290, Figures 56, 57). Additionally, the species differ from each other in valve width (15.0–19.5 μm in Chudaevia densistriata sp. nov. against 22.1 μm in Placoneis flabellata sensu Tsoy).

Table 3 Comparison of morphological features of Chudaevia densistriata sp. nov., Chudaevia flabellata comb. nov. and finds identified with these species.

	Ch. densistriata sp. nov.	Ch. densistriata sp. nov.	Ch. densistriata sp. nov.	Ch. flabellata comb. nov.	Ch. flabellata comb. nov.	N. flabellata	N. flabellata	N. flabellata	N. flabellata	N. flabellata	P. flabellata	
Note	Our material	Placoneis flabellata sensu Joh	N. diversipunctata sensu Tuji	Our material	Placoneis flabellata sensu Kimura et al.	original description from Meister	Hustedt data	Haraguchi data	Kunpradid data	sensu Moisseeva	sensu Tsoy	
Valve outline	Linear-elliptic to elliptic	Linear-elliptic to elliptic	Elliptical*	Linear-elliptic to elliptic	Oval, oval-lanceolate to broadly lanceolate	Elliptical	Elliptical*	Elliptical*	Elliptical*	Elliptical	Elliptical*	
Valve ends	Rounded to rostrate	Rounded to rostrate	Rostrate*	Rostrate	Rostrate	Rostrate*	Rostrate*	Rostrate*	Rostrate*	Weakly protracted, rostrate	Rostrate*	
Axial area	Narrow, slightly expanded to the central part of valve	Narrow, slightly expanded to the central part of valve	Narrow, slightly expanded to the central part of valve*	Narrow, slightly expanded to the central part of valve	Narrow, slightly expanded to the central part of valve	Narrow, slightly expanded to the central part of valve*	Narrow, slightly expanded to the central part of valve*	Narrow, slightly expanded to the central part of valve*	Narrow, slightly expanded to the central part of valve*	Very narrow	Narrow, slightly expanded to the central part of valve*	
Central area	Rounded to rhomboid, with single isolated stigmoid	Rhomboid, with single isolated stigmoid	Rhomboid, with single isolated stigmoid	Rhomboid, with single isolated stigmoid	Rhomboid, with single isolated stigmoid	Rhomboid, with single isolated stigmoid*	Rhomboid, with single isolated stigmoid*	Rhomboid, with single isolated stigmoid	Rhomboid, with single isolated stigmoid*	Small, rhomboid, with single isolated stigmoid	Rhomboid, with single isolated stigmoid	
Valve length, μm	25.5–64.0	34–49	42.6	24.1–57.3	(20) 26–50 (61)	27–33	38	27–33	39.7	15.0–19.5	39.3	
Valve breadth, μm	15.9–19.5	15–20	20.2	12.9–16.0	(12) 12–16	12–14	15	11–12	16.1	13–15	22.1	
Raphe	Narrow, filiform, weakly lateral towards the ends	Narrow, filiform, weakly lateral towards the ends	Narrow, filiform, weakly lateral towards the ends*	Narrow, filiform, weakly lateral towards the ends	Narrow, filiform, weakly lateral towards the ends*	Strong, straight	n.d.	Narrow, filiform, weakly lateral towards the ends*	Narrow, filiform, weakly lateral towards the ends*	n.d.	Narrow, filiform, weakly lateral towards the ends*	
Striae	Almost parallel at the central part of valve, then radiate throughout, slightly undulated. Present irregularly shortened striae on each side	Almost parallel at the central part of valve, then radiate throughout, slightly undulated. Present irregularly shortened striae on each side	Almost parallel at the central part of valve, then radiate throughout, slightly undulated. Present irregularly shortened striae on each side*	Almost parallel at the central part of valve, then radiate throughout, slightly undulated. Present irregularly shortened striae on each side	Almost parallel at the central part of valve, then radiate throughout, slightly undulated. Present irregularly shortened striae on each side*	The striae in the central part of the frustule are peculiar, reminiscent of Navicula reinhardtii Grun.	Almost parallel at the central part of valve, then radiate throughout, slightly undulated. Present irregularly shortened striae on each side*	Expressed radial striae	Almost parallel at the central part of valve, then radiate throughout, slightly undulated Present irregularly shortened striae on each side*	Slightly undulated. Present irregularly shortened striae on each side	Almost parallel at the central part of valve, then radiate throughout, slightly undulated. Present irregularly shortened striae on each side	
Striae in 10 μm	13–15	10–13	14	10–11	(8.5) 9.0–11.5 (13)	11–12	10*	11	10*	15–16	14	
Areolae in 10 μm	48–50	n.d.	n.d.	44–46	(42) 43–47 (48)	n.d.	n.d.	n.d.	n.d.	n.d.	n.d.	
Distribution	Recent. South-East Asia. Vietnam, Lake Ba Bể	East Asia. South Korea	Recent. Lake Biwa, Japan	Recent. South-East Asia. Laos	Indonesia	Recent. South-East Asia. Vietnam, Saigon River	Recent. South-East Asia. Indonesia, Musi River	East Asia. Japan, Lake Aoki	Recent. South-East Asia. Thailand, Nan Province, Nan River	Fossil. Pliocene. Primorsky Territory, Golenki village	Fossil. Early Miocene. Yamato Rise, Sea of Japan	
References	This study	Joh (2013)	Tuji (2003)	This study	Kimura, Fukushima & Kobayashi (2015)	Meister (1932)	Schmidt (1936), Hustedt (1937a)	Haraguchi (1997)	Kunpradid (2005)	Moisseeva (1971)	Tsoy (2017)	
Note:

New species are compared with similar taxa by morphology and distribution.

* counted from published data; n.d.—no data.

Comments. Morphologically similar species are known from fossil sediments. Moisseeva illustrated Navicula flabellata from Pliocene fossil sediments of Primorsky territory, Golenki village (Moisseeva, 1971: p. 145, Plate XX, figure 13). Tsoy found Placoneis flabellata in the fossil sediments from the early Miocene of Yamato upland (The Japanese Sea) (Tsoy, 2017: p. 290, Figs. 56, 57). Thus, we suggest that originally the species of this genus were widely distributed in water ecosystems across Asia, and nowadays Chudaevia gen. nov. is comprised of two recent species from Asia. Fossil specimens might represent another two species of this genus.

Chudaevia flabellata (F. Meister) Kulikovskiy, Glushchenko, Mironov & Kociolek comb. nov. (Fig. 9)

Figure 9 (A–I) Chudaevia flabellata comb. nov. Slides no. 00955 (A, D, E–I), 00996 (B, C).

(A–G) Light microscopy, size diminution series. (H, I) Scanning electron microscopy, internal view. (H) Whole valve. White arrow shows the opening of stigmoid. Black arrows show the distal raphe ends terminating on helictoglossae. (I) Note areolae with paratectula and vimines with warty outgrowths. Black arrow shows the areolar occlusion. Scale bars (A–G) 10 µm; (H) 5 µm; (I) 0.5 µm.

Basionym: Navicula flabellata Meister, 1932. Kieselalgen aus Asien. Berlin: Gebrüder Borntraeger: p. 36, fig. 94.

Synonym: Placoneis flabellata (F. Meister) Kimura, Fukushima & Kobayashi, 2015.

Description. LM (Figs. 9A–9G). The structure of chloroplast is studied in Kimura, Fukushima & Kobayashi (2015). Valves linear-elliptic to elliptic. Valve ends acutely rounded, barely protracted, rostrate. Length 24.1–57.3 µm, breadth 12.9–16.0 µm. Axial area narrow, slightly expanded to the central part of valve. Central area rhomboid. In the center of valves are present irregularly shortened striae on each side. Raphe narrow, filiform, weakly lateral towards the ends. Proximal raphe ends drop-shaped. Distal raphe ends extend to valve margin. One stigmoid located closely to the central raphe ends, slightly offset laterally relative to the axial area; at the post-initial valves is weakly expressed. Striae almost parallel at the central part of valve, then radiate throughout, slightly undulated, 10–11 in 10 µm. Areolae not resolvable in LM.

SEM, internal view (Figs. 9H, 9I). The raphe is straight, proximal raphe ends slightly deflected to one, towards the stigmoid. Distal ends terminating on small helictoglossae and slightly deflected to opposite sides from proximal raphe ends (Fig. 9H, black arrows). Stigmoid opening elongated (Fig. 9H, white arrow). Internal opening of the stigmoid is noticeable in LM with careful focusing. Areolae in striae are apically elongated, striae separated by wide interstriae (virgae). Warty outgrowths are present between the areolae, at the vimines. Areolae elliptical with 2–4 projections (struts) extending into the lumen of the openings, thus forming a paratectulum (Fig. 9I, black arrow). Areolae 44–46 in 10 µm.

Distribution. South-East Asia: as Navicula flabellata, the species was illustrated from Vietnam, Saigon River (Meister, 1932); Indonesia, Musi River, Palembang (Schmidt, 1936; Hustedt, 1937b); Thailand, Nan Province, Nan River (Kunpradid, 2005). East Asia: Japan, Lake Aoki (Haraguchi, 1997). As Placoneis flabellata, the species was illustrated from Indonesia, unnamed river in Batimurung, Sulawesi Island (Kimura, Fukushima & Kobayashi, 2015). We illustrate this species from Laos (Vientiane Province, periphyton of Nam Lik River; Khammouane Province, unnamed creek, Nahin village).

Comments. The original description of N. flabellata was proposed by Meister (1932) based on M. Voigt’s samples from Vietnam. In that description, the presence of stigmoid is not mentioned. In spite of that, the stigmoid is resolvable (Meister, 1932: Figure 94). Later, Hustedt highlighted the presence of stigmoid in the species from Indonesia (Schmidt, 1936; Hustedt, 1937b: Taf. XVIII, Figure 17).

New combinations

Witkowskia neohambergii (Kezlya, Glushchenko, Kulikovskiy & Kociolek) Kulikovskiy, Glushchenko, Mironov & Kociolek comb. nov.

Basionym: Placoneis neohambergii Kezlya, Glushchenko, Kulikovskiy & Kociolek in Kezlya et al., 2021. Three New Species of Placoneis Mereschkowsky (Bacillariophyceae: Cymbellales) with Comments on Cryptic Diversity in the P. elginensis—Group. Water 2021, 13, p. 8, figs. 8, 9.

Witkowskia abiskoensis (Hustedt) Kulikovskiy, Glushchenko, Mironov & Kociolek comb. nov.

Basionym: Navicula abiskoensis Hustedt, 1942a. Diatomeen aus der Umgebung von Abisko in Schwedisch-Lappland. Archiv für Hydrobiologie 39(1), p. 118, fig. 36.

Synonym: Placoneis abiskoensis (Hustedt) Lange-Bertalot & Metzeltin in Metzeltin & Witkowski, 1996.

Witkowskia abundans (Metzeltin, Lange-Bertalot & García-Rodríguez) Kulikovskiy, Glushchenko, Mironov & Kociolek comb. nov.

Basionym: Placoneis abundans Metzeltin, Lange-Bertalot & García-Rodriguez, 2005. Diatoms of Uruguay. Compared with other taxa from South America and elsewhere. Iconographia Diatomologica 15, p. 166, pl. 73, figs. 1–14; pl. 77, figs. 1–3A.

Witkowskia abyssalis (Pomazkina & Sherbakova in Pomazkina, Rodionova & Sherbakova) Kulikovskiy, Glushchenko, Mironov & Kociolek comb. nov.

Basionym: Placoneis abyssalis Pomazkina & Sherbakova 2019 in Pomazkina, Rodionova & Sherbakova, 2019. Validation of 123 names of new diatom taxa from Lake Baikal. Limnology and Freshwater Biology 2019(1), p. 192.

Synonym: Placoneis abyssalis Pomazkina, Rodionova & Sherbakova, 2018, nom. invalid.

Witkowskia acuta (Pomazkina & Rodionova in Pomazkina, Rodionova & Sherbakova) Kulikovskiy, Glushchenko, Mironov & Kociolek comb. nov.

Basionym: Placoneis acuta Pomazkina & Rodionova in Pomazkina, Rodionova & Sherbakova, 2019. Validation of 123 names of new diatom taxa from Lake Baikal. Limnology and Freshwater Biology 2019(1), p. 192.

Synonym: Placoneis acuta Pomazkina & Rodionova in Pomazkina, Rodionova & Sherbakova, 2018, nom. invalid.

Witkowskia amoena (Metzeltin, Kulikovskiy & Lange-Bertalot in Kulikovskiy, Lange-Bertalot, Metzeltin & Witkowski) Kulikovskiy, Glushchenko, Mironov & Kociolek comb. nov.

Basionym: Placoneis amoena Metzeltin, Kulikovskiy & Lange-Bertalot in Kulikovskiy et al., 2012. Lake Baikal: Hotspot of endemic diatoms I. Iconographia Diatomologica 23, p. 224, pl. 124: fig. 1.

Witkowskia anglophila (Lange-Bertalot in Lange-Bertalot & Krammer) Kulikovskiy, Glushchenko, Mironov & Kociolek comb. nov.

Basionym: Navicula anglophila Lange-Bertalot in Lange-Bertalot & Krammer, 1987. Bacillariaceae, Epithemiaceae, Surirellaceae. Neue und wenig bekannte Taxa, neue Kombinationen und Synonyme sowie Bemerkungen und Ergänzungen zu den Naviculaceae. Bibliotheca Diatomologica 15: p. 121.

Witkowskia antediluviana (Hustedt) Kulikovskiy, Glushchenko, Mironov & Kociolek comb. nov.

Basionym: Navicula antediluviana Hustedt, 1955a. Neue und wenig bekannte Diatomeen. VII. Berircht der Deutschen Botanischen Gessellschaft 68(3), p. 132, figs 2, 3.

Synonym: Placoneis antediluviana (Hustedt) Lange-Bertalot in Metzeltin, Lange-Bertalot & García-Rodriguez, 2005.

Witkowskia apicalicostata (Metzeltin & Lange-Bertalot) Kulikovskiy, Glushchenko, Mironov & Kociolek comb. nov.

Basionym: Placoneis apicalicostata Metzeltin & Lange-Bertalot, 2002. Diatoms from the “Island Continent” Madagascar. Iconographia Diatomologica 11, p. 53, pl. 30, figs 1–7.

Witkowskia argentata (Pomazkina & Sherbakova in Pomazkina, Rodionova & Sherbakova) Kulikovskiy, Glushchenko, Mironov & Kociolek comb. nov.

Basionym: Placoneis argentata Pomazkina & Sherbakova in Pomazkina, Rodionova & Sherbakova, 2019. Validation of 123 names of new diatom taxa from Lake Baikal. Limnology and Freshwater Biology 2019(1), p. 193.

Synonym: Placoneis argentata Pomazkina & Sherbakova in Pomazkina, Rodionova & Sherbakova, 2018, nom. invalid.

Witkowskia asymmetricus (Glushchenko, Keslya, Kulikovskiy & Kociolek) Kulikovskiy, Glushchenko, Mironov & Kociolek comb. nov.

Basionym: Placoneis asymmetricus Glushchenko, Keslya, Kulikovskiy & Kociolek in Kezlya et al., 2022. A new species of Placoneis Mereschkowsky (Bacillariophyceae: Cymbellales) from wet soils in southern Vietnam. Cryptogamie, Algologie 43(11), p. 180, figs 2–4.

Witkowskia attenuata (Pomazkina & Rodionova in Pomazkina, Rodionova & Sherbakova) Kulikovskiy, Glushchenko, Mironov & Kociolek comb. nov.

Basionym: Placoneis attenuata Pomazkina & Rodionova in Pomazkina, Rodionova & Sherbakova, 2019. Validation of 123 names of new diatom taxa from Lake Baikal. Limnology and Freshwater Biology 2019(1), p. 193.

Synonym: Placoneis attenuata Pomazkina & Rodionova in Pomazkina, Rodionova & Sherbakova, 2018, nom. invalid.

Witkowskia australis (Van de Vijver & Zidarova in Zidarova, Van de Vijver, Mataloni, Kopalova & Nedbalova) Kulikovskiy, Glushchenko, Mironov & Kociolek comb. nov.

Basionym: Placoneis australis Van de Vijver & Zidarova in Zidarova et al., 2009. Four new freshwater diatom species (Bacillariophyceae) from Antarctica. Cryptogamie, Algologie 30(4), p. 301, figs 44–58, figs 62–64.

Witkowskia baicalensis (Pomazkina & Sherbakova in Pomazkina, Rodionova & Sherbakova) Kulikovskiy, Glushchenko, Mironov & Kociolek comb. et stat. nov.

Basionym: Placoneis witkowskii var. baicalensis Pomazkina & Sherbakova in Pomazkina, Rodionova & Sherbakova, 2019. Validation of 123 names of new diatom taxa from Lake Baikal. Limnology and Freshwater Biology 2019(1), p. 198.

Synonym: Placoneis witkowskii var. baicalensis Pomazkina & Sherbakova in Pomazkina, Rodionova & Sherbakova, 2018, nom. invalid.

Witkowskia baikaloelginensis (Kezlya, Glushchenko, Kulikovskiy & Kociolek) Kulikovskiy, Glushchenko, Mironov & Kociolek comb. nov.

Basionym: Placoneis baikaloelginensis Kezlya, Glushchenko, Kulikovskiy & Kociolek in Kezlya et al., 2021. Three New Species of Placoneis Mereschkowsky (Bacillariophyceae: Cymbellales) with Comments on Cryptic Diversity in the P. elginensis—Group. Water 2021, 13, p. 4, figs. 2–4.

Witkowskia betulina (Pomazkina & Rodionova in Pomazkina, Rodionova & Sherbakova) Kulikovskiy, Glushchenko, Mironov & Kociolek comb. nov.

Basionym: Placoneis betulina Pomazkina & Rodionova in Pomazkina, Rodionova & Sherbakova, 2019. Validation of 123 names of new diatom taxa from Lake Baikal. Limnology and Freshwater Biology 2019(1), p. 193.

Synonym: Placoneis betulina Pomazkina & Rodionova in Pomazkina, Rodionova & Sherbakova, 2018, nom. invalid.

Witkowskia bicapitata (Heinzerling) Kulikovskiy, Glushchenko, Mironov & Kociolek comb. nov.

Basionym: Placoneis bicapitata Heinzerling, 1908. Der Bau der Diatomeenzelle mit besonderer Beruchsichtigung der ergastischen Gebilde und der Beziegung des Baues zur Systematic. Bibliotheca Botanica. Stuttgart 15(69), p. 71, pl. I, fig. 28.

Witkowskia bicuneus (Metzeltin, Lange-Bertalot & García-Rodríguez) Kulikovskiy, Glushchenko, Mironov & Kociolek comb. nov.

Basionym: Placoneis bicuneus Metzeltin, Lange-Bertalot & García-Rodriguez, 2005. Diatoms of Uruguay. Compared with other taxa from South America and elsewhere. Iconographia Diatomologica 15, p. 170, pl. 71, figs. 1–7, pl. 76, fig. 1.

Witkowskia bona (Pomazkina & Sherbakova in Pomazkina, Rodionova & Sherbakova) Kulikovskiy, Glushchenko, Mironov & Kociolek comb. nov.

Basionym: Placoneis bona Pomazkina & Sherbakova in Pomazkina, Rodionova & Sherbakova, 2019. Validation of 123 names of new diatom taxa from Lake Baikal. Limnology and Freshwater Biology 2019(1), p. 193.

Synonym: Placoneis bona Pomazkina & Sherbakova in Pomazkina, Rodionova & Sherbakova, 2018, nom invalid.

Witkowskia boris-skvortzowii (Metzeltin, Kulikovskiy & Lange-Bertalot in Kulikovskiy, Lange-Bertalot, Metzeltin & Witkowski) Kulikovskiy, Glushchenko, Mironov & Kociolek comb. nov.

Basionym: Placoneis boris-skvortzowii Metzeltin, Kulikovskiy & Lange-Bertalot in Kulikovskiy et al., 2012. Lake Baikal: Hotspot of endemic diatoms I. Iconographia Diatomologica 23, p. 225, pl. 132, figs. 17, 18.

Witkowskia bukhchuluunae (Metzeltin, Lange-Bertalot & Nergui) Kulikovskiy, Glushchenko, Mironov & Kociolek comb. nov.

Basionym: Placoneis bukhchuluunae Metzeltin, Lange-Bertalot & Soninkhishig, 2009. Diatoms in Mongolia. Iconographia Diatomologica 20, p. 81, pl. 55, figs. 7–13.

Witkowskia capitata (Patrick) Kulikovskiy, Glushchenko, Mironov & Kociolek comb. et stat. nov.

Basionym: Navicula exigua var. capitata Patrick, 1945. A taxonomic and ecological study of some diatoms from the Pocono Plateau and adjacent regions. Farlowia. 2(2), p. 179, pl. 1, fig. 8.

Synonym: Placoneis exigua var. capitata (Patrick) Aysel, 2005.

Witkowskia cattiensis (Glushchenko, Kezlya, Kulikovskiy & Kociolek) Kulikovskiy, Glushchenko, Mironov & Kociolek comb. nov.

Basionym: Placoneis cattiensis Glushchenko, Kezlya, Kulikovskiy & Kociolek in Glushchenko et al., 2020. Placoneis cattiensis sp. nov.–a new diatom (Bacillariophyceae: Cymbellales) soil species from Cát Tiên National Park (Vietnam). Phytotaxa 460(4): p. 241, figs. 3–28.

Witkowskia centropunctata (Hustedt) Kulikovskiy, Glushchenko, Mironov & Kociolek comb. nov.

Basionym: Navicula centropunctata Hustedt, 1964. Die Kieselalgen Deutschlands, Österreichs und der Schweiz unter Berücksichtigung der übrigen Länder Europas sowie der angrenzenden Meeresgebiete. In: L. Rabenhorst (ed.), Kryptogamen Flora von Deutschland, Österreich und der Schweiz. Akademische Verlagsgesellschaft m.b.h. Leipzig 7(Teil 3, Lief. 4), p. 677, fig. 1678.

Synonym: Placoneis centropunctata (Hustedt) Metzeltin & Lange-Bertalot, 1998.

Witkowskia chilensis (Lange-Bertalot & U. Rumrich in U. Rumrich, Lange-Bertalot & M. Rumrich) Kulikovskiy, Glushchenko, Mironov & Kociolek comb. nov.

Basionym: Placoneis chilensis Lange-Bertalot & U. Rumrich in Rumrich, Lange-Bertalot & Rumrich, 2000. Diatoms of the Andes. From Venezuela to Patagonia/Tierra del Fuego and two additional contributions. Iconographia Diatomologica 9, p. 206, pl. 59, figs. 8, 9.

Witkowskia clementioides (Hustedt) Kulikovskiy, Glushchenko, Mironov & Kociolek comb. nov.

Basionym: Navicula clementioides Hustedt, 1944. Neue und wenig bekannte Diatomeen. Bericht der Deutschen Botanischen Gessellschaft 61, p. 285, pl. 8, figs. 19, 20.

Synonym: Placoneis clementioides (Hustedt) Cox, 1987.

Witkowskia clementis (Grunow) Kulikovskiy, Glushchenko, Mironov & Kociolek comb. nov.

Basionym: Navicula clementis Grunow, 1882. Beiträge zur Kenntniss der fossilen Diatomeen Österreich-Ungarns. In: Beiträge zur Paläontologie Österreich-Ungarns und des Orients. II Band Pt 4. (Mojsisovics, E. & Neumayr, N. Eds). Wein: Alfred Hölder, p. 144, pl. 30, fig. 52.

Synonym: Placoneis clementis (Grunow) Cox, 1987.

Witkowskia clementispronina (Lange-Bertalot & Wojtal) Kulikovskiy, Glushchenko, Mironov & Kociolek comb. nov.

Basionym: Placoneis clementispronina Lange-Bertalot & Wojtal, 2014. Diversity in species complexes of Placoneis clementis (Grunow) Cox and Paraplaconeis placentula (Ehrenberg) Kulikovskiy, Lange-Bertalot & Metzeltin. Beihefte zur Nova Hedwigia 143, p. 405, fig. 5.

Witkowskia cocquytiae (Fofana, Sow, Taylor, Ector & Van de Vijver) Kulikovskiy, Glushchenko, Mironov & Kociolek comb. nov.

Basionym: Placoneis cocquytiae Fofana et al., 2014. Placoneis cocquytiae, a new raphid diatom (Bacillariophyceae) from the Senegal River (Senegal, West Africa). Phytotaxa 161(2), p. 140, figs. 1–10.

Witkowskia composita (Pomazkina & Sherbakova in Pomazkina, Rodionova & Sherbakova) Kulikovskiy, Glushchenko, Mironov & Kociolek comb. nov.

Basionym: Placoneis composita Pomazkina & Sherbakova in Pomazkina, Rodionova & Sherbakova, 2019. Validation of 123 names of new diatom taxa from Lake Baikal. Limnology and Freshwater Biology, p. 193.

Synonym: Placoneis composita Pomazkina & Sherbakova in Pomazkina, Rodionova & Sherbakova, 2018, nom. invalid.

Witkowskia constans (Hustedt) Kulikovskiy, Glushchenko, Mironov & Kociolek comb. nov.

Basionym: Navicula constans Hustedt, 1944. Neue und wenig bekannte Diatomeen. Bericht der Deutschen Botanischen Gessellschaft 61, P. 284, pl. VIII, fig. 13.

Synonym: Placoneis constans (Hustedt) Cox, 2003.

Witkowskia conveniens (Hustedt) Kulikovskiy, Glushchenko, Mironov & Kociolek comb. nov.

Basionym: Navicula conveniens Hustedt, 1952. Neue und wenig bekannte Diatomeen. IV. Botaniska Notiser, p. 402, fig. 123.

Synonym: Placoneis conveniens (Hustedt) Metzeltin & Lange-Bertalot, 1998.

Witkowskia coxiae (Kociolek & Thomas) Kulikovskiy, Glushchenko, Mironov & Kociolek comb. nov.

Basionym: Placoneis coxiae Kociolek & Thomas, 2010. Taxonomy and ultrastructure of five naviculoid diatoms (class Bacillariophyceae) from the Rocky Mountains of Colorado (USA), with the description of a new genus and four new species. Nova Hedwigia 90(1/2), p. 200, figs. 23–29, figs. 36–38.

Witkowskia cruciata (Pomazkina & Rodionova in Pomazkina, Rodionova & Sherbakova) Kulikovskiy, Glushchenko, Mironov & Kociolek comb. nov.

Basionym: Placoneis cruciata Pomazkina & Rodionova in Pomazkina, Rodionova & Sherbakova, 2019. Validation of 123 names of new diatom taxa from Lake Baikal. Limnology and Freshwater Biology 2019(1), p. 193.

Synonym: Placoneis cruciata Pomazkina & Rodionova in Pomazkina, Rodionova & Sherbakova, 2018, nom. invalid.

Witkowskia cuneata (M. Möller ex Foged) Kulikovskiy, Glushchenko, Mironov & Kociolek comb. et stat. nov.

Basionym: Navicula dicephala f. cuneata Foged, 1977. Freshwater diatoms in Ireland. Bibliotheca Phycologica 34, p. 78, pl. XXIX, fig. 6 (as ‘(M. Møller) fo. Nov.’).

Synonyms: Placoneis elginensis var. cuneata (M. Møller ex Foged) Lange-Bertalot in Krammer & Lange-Bertalot, 1985; Placoneis cuneata (M. Möller ex Foged) Potapova, 2014.

Witkowskia dacostae (Metzeltin & Lange-Bertalot) Kulikovskiy, Glushchenko, Mironov & Kociolek comb. nov.

Basionym: Placoneis dacostae Metzeltin & Lange-Bertalot, 1998. Tropical diatoms of South America I: About 700 predominantly rarely known or new taxa representative of the neotropical flora. Iconographia Diatomologica 5, p. 196, pl. 77, fig. 1, pl. 90, figs. 4–8.

Witkowskia dahurica (Skvortzow) Kulikovskiy, Glushchenko, Mironov & Kociolek comb. nov.

Basionym: Navicula dahurica Skvortzow, 1937. Bottom diatoms from Olhon Gate of Baikal Lake, Siberia. Philippine Journal of Science, Section C 62(3), p. 337, pl. 7, fig. 35; pl. 8, fig. 7.

Synonym: Placoneis dahurica (Skvortzow) Pomazkina & Rodionova in Pomazkina, Rodionova & Sherbakova, 2019.

Witkowskia demeraroides (Hustedt) Kulikovskiy, Glushchenko, Mironov & Kociolek comb. nov.

Basionym: Navicula demeraroides Hustedt, 1964. Die Kieselalgen Deutschlands, Österreichs und der Schweiz unter Berücksichtigung der übrigen Länder Europas sowie der angrenzenden Meeresgebiete. In: L. Rabenhorst (ed.), Kryptogamen Flora von Deutschland, Österreich und der Schweiz. Akademische Verlagsgesellschaft m.b.h. Leipzig 7 (Teil 3, Lief. 4), p. 676, fig. 1677.

Synonym: Placoneis demeraroides (Hustedt) Metzeltin & Lange-Bertalot, 1998.

Witkowskia densa (Hustedt) Kulikovskiy, Glushchenko, Mironov & Kociolek comb. nov.

Basionym: Navicula densa Hustedt, 1944. Neue und wenig bekannte Diatomeen. Bericht der Deutschen Botanischen Gessellschaft 61, p. 284, fig. 28.

Synonym: Placoneis densa (Hustedt) Lange-Bertalot in Metzeltin, Lange-Bertalot & García-Rodriguez, 2005.

Witkowskia diaphana (Pomazkina & Rodionova in Pomazkina, Rodionova & Sherbakova) Kulikovskiy, Glushchenko, Mironov & Kociolek comb. nov.

Basionym: Placoneis diaphana Pomazkina & Rodionova in Pomazkina, Rodionova & Sherbakova, 2019. Validation of 123 names of new diatom taxa from Lake Baikal. Limnology and Freshwater Biology 2019(1), p. 194.

Synonym: Placoneis diaphana Pomazkina & Rodionova in Pomazkina, Rodionova & Sherbakova, 2018, nom. invalid.

Witkowskia dicephala (Ehrenberg) Kulikovskiy, Glushchenko, Mironov & Kociolek comb. nov.

Basionym: Navicula dicephala Ehrenberg, 1838. Die Infusionsthierchen als vollkommene Organismen. Ein Blick in das tiefere organische Leben der Natur. P. 185, no figures.

Synonym: Placoneis dicephala (Ehrenberg) Mereschkowsky, 1903.

Witkowskia disparilis (Hustedt) Kulikovskiy, Glushchenko, Mironov & Kociolek comb. nov.

Basionym: Placoneis disparilis (Hustedt) Metzeltin & Lange-Bertalot, 1998. Tropical diatoms of South America I: About 700 predominantly rarely known or new taxa representative of the neotropical flora. Iconographia Diatomologica 5, p. 197, pl. 92, figs. 1–6.

Witkowskia distincta (Pomazkina & Sherbakova in Pomazkina, Rodionova & Sherbakova) Kulikovskiy, Glushchenko, Mironov & Kociolek comb. nov.

Basionym: Placoneis distincta Pomazkina & Sherbakova in Pomazkina, Rodionova & Sherbakova, 2019. Validation of 123 names of new diatom taxa from Lake Baikal. Limnology and Freshwater Biology 2019(1), p. 194.

Synonym: Placoneis distincta Pomazkina & Sherbakova in Pomazkina, Rodionova & Sherbakova, 2018, nom. invalid.

Witkowskia diversipunctata (Hustedt) Kulikovskiy, Glushchenko, Mironov & Kociolek comb. nov

Basionym: Navicula diversipunctata Hustedt, 1944. Neue und wenig bekannte Diatomeen. Bericht der Deutschen Botanischen Gessellschaft 61, p. 275, fig. 5.

Witkowskia diminuta (Pomazkina in Pomazkina, Rodionova & Sherbakova) Kulikovskiy, Glushchenko, Mironov & Kociolek comb. et stat. nov.

Basionym: Placoneis vladimiri var. diminuta Pomazkina in Pomazkina, Rodionova & Sherbakova, 2019. Validation of 123 names of new diatom taxa from Lake Baikal. Limnology and Freshwater Biology 2019(1), p. 198.

Synonym: Placoneis vladimiri var. diminuta Pomazkina in Pomazkina, Rodionova & Sherbakova, 2018, nom. invalid.

Witkowskia edlundii (Vishnyakov, Kulikovskiy, Dorofeyuk & Genkal) Kulikovskiy, Glushchenko, Mironov & Kociolek comb. nov.

Basionym: Placoneis edlundii Vishnyakov in Vishnyakov et al., 2016. New species and new combinations in the genera Placoneis and Paraplaconeis (Bacillariophyceae: Cymbellales). Botanicheskii Zhurnal, p. 1302, pl. I, figs. 13–25; pl. II, fig. 7.

Witkowskia elegans (Metzeltin, Lange-Bertalot & García-Rodríguez) Kulikovskiy, Glushchenko, Mironov & Kociolek comb. nov.

Basionym: Placoneis elegans Metzeltin, Lange-Bertalot & García-Rodriguez, 2005. Diatoms of Uruguay. Compared with other taxa from South America and elsewhere. Iconographia Diatomologica 5, p. 174, pl. 75, figs. 1–10.

Witkowskia elegantula (Metzeltin, Lange-Bertalot & García-Rodríguez) Kulikovskiy, Glushchenko, Mironov & Kociolek comb. nov.

Basionym: Placoneis elegantula Metzeltin, Lange-Bertalot & García-Rodriguez, 2005. Diatoms of Uruguay. Compared with other taxa from South America and elsewhere. Iconographia Diatomologica 5, p. 74, pl. 75, figs. 1–10.

Witkowskia elenae (Pomazkina & Sherbakova in Pomazkina, Rodionova & Sherbakova) Kulikovskiy, Glushchenko, Mironov & Kociolek comb. nov.

Basionym: Placoneis elenae Pomazkina & Sherbakova in Pomazkina, Rodionova & Sherbakova, 2019. Validation of 123 names of new diatom taxa from Lake Baikal. Limnology and Freshwater Biology 2019(1), p. 194.

Synonym: Placoneis elenae Pomazkina & Sherbakova in Pomazkina, Rodionova & Sherbakova, 2018, nom invalid.

Witkowskia elginensis (Gregory) Kulikovskiy, Glushchenko, Mironov & Kociolek comb. nov.

Basionym: Pinnularia elginensis Gregory, 1856. Notice of some new species of British Fresh-water Diatomaceae. Quarterly Journal of Microscopical Science, new series, London 4, p. 9, pl. 1, fig. 33.

Synonym: Placoneis elginensis (Gregory) Cox, 1987.

Witkowskia elliptica (Hustedt) Kulikovskiy, Glushchenko, Mironov & Kociolek comb. et stat. nov.

Basionym: Navicula exigua var. elliptica Hustedt, 1927. Fossile bacillariaceen aus dem Loa-Becken in der Atacama-Wüste, Chile. Archiv für Hydrobiologie 18(2), p. 244, pl. 7, fig. 27.

Synonym: Placoneis elliptica (Hustedt) Ohtsuka, 2002.

Witkowskia ellipticorostrata (Metzeltin, Lange-Bertalot & Nergui) Kulikovskiy, Glushchenko, Mironov & Kociolek comb. nov.

Basionym: Placoneis ellipticorostrata Metzeltin, Lange-Bertalot & Soninkhishig, 2009. Diatoms in Mongolia. Iconographia Diatomologica 20, p. 82, pl. 53, figs. 1–5.

Witkowskia eugeniae (Sherbakova in Pomazkina, Rodionova & Sherbakova) Kulikovskiy, Glushchenko, Mironov & Kociolek comb. nov.

Basionym: Placoneis eugeniae Sherbakova in Pomazkina, Rodionova & Sherbakova, 2019. Validation of 123 names of new diatom taxa from Lake Baikal. Limnology and Freshwater Biology 2019(1), p. 194.

Synonym: Placoneis eugeniae Sherbakova in Pomazkina, Rodionova & Sherbakova, 2019, nom. invalid.

Witkowskia exigua (W. Gregory) Kulikovskiy, Glushchenko, Mironov & Kociolek comb. nov.

Basionym: Pinnularia exigua Gregory, 1854. Notice of the new forms and varieties of known forms occurring in the diatomaceous earth of Mull; with remarks on the classification of the Diatomaceae. Quarterly Journal of Microscopical Science 2, p. 99, pl. IV, fig. 14.

Synonym: Placoneis exigua (W. Gregory) Mereschkowsky, 1903.

Witkowskia exiguiformis (Hustedt) Kulikovskiy, Glushchenko, Mironov & Kociolek comb. nov.

Basionym: Navicula exiguiformis Hustedt, 1944. Neue und wenig bekannte Diatomeen. Bericht der Deutschen Botanischen Gessellschaft 61, p. 283, fig. 23.

Synonym: Placoneis exiguiformis (Hustedt) Lange-Bertalot in Metzeltin, Lange-Bertalot & García-Rodriguez, 2005.

Witkowskia exiguioides (Hustedt) Kulikovskiy, Glushchenko, Mironov & Kociolek comb. nov.

Basionym: Navicula exiguioides Hustedt, 1955a. Neue und wenig bekannte Diatomeen. VII. Bericht der Deutschen Botanischen Gessellschaft 68(3), p. 131, fig. 1.

Synonym: Placoneis exiguioides (Hustedt) Lange-Bertalot in Metzeltin, Lange-Bertalot & García-Rodriguez, 2005.

Witkowskia explanata (Hustedt) Kulikovskiy, Glushchenko, Mironov & Kociolek comb. nov.

Basionym: Navicula explanata Hustedt, 1948. Die Diatomeenflora diluvialer Sedimente bei dem Dorfe Gaj bei Konin im Warthegebiet. Schweizerische Zeitschrift für Hydrologie 11(1/2), p. 202, 207, figs. 7, 8.

Synonyms: Placoneis explanata (Hustedt) S. Mayama in Mayama & Kawashima, 1998; Placoneis explanata (Hustedt) Lange-Bertalot in Rumrich, Lange-Bertalot & Rumrich, 2000.

Witkowskia extraordinaris (Pomazkina & Rodionova in Pomazkina, Rodionova & Sherbakova) Kulikovskiy, Glushchenko, Mironov & Kociolek comb. et stat. nov.

Basionym: Placoneis extraordinaris Pomazkina & Rodionova in Pomazkina, Rodionova & Sherbakova, 2019. Validation of 123 names of new diatom taxa from Lake Baikal. Limnology and Freshwater Biology 2019(1), p. 194.

Synonym: Placoneis extraordinaris Pomazkina & Rodionova in Pomazkina, Rodionova & Sherbakova, 2018, nom. invalid.

Witkowskia fogedii (Foged & Møller) Kulikovskiy, Glushchenko, Mironov & Kociolek comb. et nom. nov.

Replaced synonym: Navicula pseudoanglica Foged & Møller in Foged, 1968. Diatoméerne I en postglacial boreprøve fra bunden af Esrom sø, Danmark. Medd. Dansk Geol. Foren. København., Bd 18: 178, fig. 1 in appendix, nom. invalid.

Synonym: Placoneis incerta Vishnyakov et al., 2016.

Witkowskia fourtanieri (Kociolek & Thomas) Kulikovskiy, Glushchenko, Mironov & Kociolek comb. nov.

Basionym: Placoneis fourtanieri Kociolek & Thomas, 2010. Taxonomy and ultrastructure of five naviculoid diatoms (class Bacillariophyceae) from the Rocky Mountains of Colorado (USA), with the description of a new genus and four new species. Nova Hedwigia 90 (1–2), p. 204, figs. 42–47, 52–56.

Witkowskia gastriformis (Hustedt) Kulikovskiy, Glushchenko, Mironov & Kociolek comb. nov.

Basionym: Navicula gastriformis Hustedt, 1935. Die fossile Diatomeenflora in den Ablagerungen des Tobasees auf Sumatra. “Tropische Binnengewasser, Band V”. Archiv für Hydrobiologie, Supplement 14, p. 157, pl. 1, fig. 8.

Synonyms: Navicula gastriformis Hustedt in Schmidt, 1934, nom. invalid; Placoneis gastriformis (Hustedt) Lange-Bertalot in Metzeltin, Lange-Bertalot & García-Rodriguez, 2005.

Witkowskia geitleri (Hustedt) Kulikovskiy, Glushchenko, Mironov & Kociolek comb. nov.

Basionym: Navicula geitleri Hustedt, 1937a. Systematische und ökologische Untersuchungen über die Diatomeen-Flora von Java, Bali und Sumatra nach dem Material der Deutschen Limnologischen Sunda-Expedition. Archiv für Hydrobiologie (Supplement) 15(2), p. 263, figs. 1–3; pl. XVIII, fig. 34.

Synonyms: Navicula geitleri Hustedt in A. Schmidt, 1934, nom. invalid.; Placoneis geitleri (Hustedt) Vishnjakov in Vishnyakov et al., 2016.

Witkowskia gelegma (Kulikovskiy, Lange-Bertalot & Metzeltin in Kulikovskiy, Lange-Bertalot, Metzeltin & Witkowski) Kulikovskiy, Glushchenko, Mironov & Kociolek comb. nov.

Basionym: Placoneis gelegma Kulikovskiy, Lange-Bertalot & Metzeltin in Kulikovskiy et al., 2012. Lake Baikal: Hotspot of endemic diatoms I. Iconographia Diatomologica 23, p. 226, pl. 132, figs. 19–21.

Witkowskia gracilis (Metzeltin, Lange-Bertalot & García-Rodríguez) Kulikovskiy, Glushchenko, Mironov & Kociolek comb. nov.

Basionym: Placoneis gracilis Metzeltin, Lange-Bertalot & García-Rodriguez, 2005. Diatoms of Uruguay. Compared with other taxa from South America and elsewhere. Iconographia Diatomologica 15, p. 180, pl. 73, figs. 15–17; pl. 76, fig. 3.

Witkowskia granum (Pomazkina & Sherbakova in Pomazkina, Rodionova & Sherbakova) Kulikovskiy, Glushchenko, Mironov & Kociolek comb. nov.

Basionym: Placoneis granum Pomazkina & Sherbakova in Pomazkina, Rodionova & Sherbakova, 2019. Validation of 123 names of new diatom taxa from Lake Baikal. Limnology and Freshwater Biology 2019(1), p. 194.

Synonym: Placoneis granum Pomazkina & Sherbakova in Pomazkina, Rodionova & Sherbakova, 2018, nom. invalid.

Witkowskia grata (Pomazkina & Sherbakova in Pomazkina, Rodionova & Sherbakova) Kulikovskiy, Glushchenko, Mironov & Kociolek comb. nov.

Basionym: Placoneis grata Pomazkina in Pomazkina, Rodionova & Sherbakova, 2019. Validation of 123 names of new diatom taxa from Lake Baikal. Limnology and Freshwater Biology 2019(1), p. 195.

Synonym: Placoneis grata Pomazkina in Pomazkina, Rodionova & Sherbakova, 2018, nom. invalid.

Witkowskia habita (Hustedt) Kulikovskiy, Glushchenko, Mironov & Kociolek comb. nov.

Basionym: Navicula habita Hustedt, 1952. Neue und wenig bekannte Diatomeen. IV. Botaniska Notiser, p. 399, fig. 116.

Synonym: Placoneis habita (Hustedt) Lange-Bertalot in Metzeltin, Lange-Bertalot & García-Rodriguez, 2005.

Witkowskia hambergii (Hustedt) Kulikovskiy, Glushchenko, Mironov & Kociolek comb. nov.

Basionym: Navicula hambergii Hustedt, 1924. Die Bacillariaceen-Vegetation des Sarekgebirges. Naturwissenschaftliche Untersuchungen des Sarekgebirges in Schwedisch-Lappland, Botanik, Stockholm 3(6), p. 562, pl. 17, fig. 2.

Synonym: Placoneis hambergii (Hustedt) Bruder in Bruder & Medlin, 2007.

Witkowskia humilis (Metzeltin, Lange-Bertalot & García-Rodríguez) Kulikovskiy, Glushchenko, Mironov & Kociolek comb. nov.

Basionym: Placoneis humilis Metzeltin, Lange-Bertalot & García-Rodriguez, 2005. Diatoms of Uruguay. Compared with other taxa from South America and elsewhere. Iconographia Diatomologica 15, p. 182, pl. 74, figs. 11–19; pl. 76, fig. 4.

Witkowskia hustedtii (Hustedt) Kulikovskiy, Glushchenko, Mironov & Kociolek comb. nov., stat. nov. et nom. nov.

Replaced synonym: Navicula exigua var. signata Hustedt, 1944. New and little known Diatoms. Report of the German Botanical Society 61, p. 287, fig. 14.

Synonym: Placoneis significans Lange-Bertalot in Metzeltin, Lange-Bertalot & García-Rodriguez, 2005.

Witkowskia ignita (Pomazkina & Sherbakova in Pomazkina, Rodionova & Sherbakova) Kulikovskiy, Glushchenko, Mironov & Kociolek comb. nov.

Basionym: Placoneis ignita Pomazkina & Sherbakova in Pomazkina, Rodionova & Sherbakova, 2019. Validation of 123 names of new diatom taxa from Lake Baikal. Limnology and Freshwater Biology 2019(1), p. 195.

Synonym: Placoneis ignita Pomazkina & Sherbakova in Pomazkina, Rodionova & Sherbakova, 2018, nom invalid.

Witkowskia ignorata (Schimanski) Kulikovskiy, Glushchenko, Mironov & Kociolek comb. nov.

Basionym: Navicula ignorata Schimanski, 1978. Contribution to the diatom flora of the Franconian Forest. Nova Hedwigia 30, p. 585, pl. 6, figs. 1–9.

Synonym: Placoneis ignorata (Schimanski) Lange-Bertalot in Rumrich, Lange-Bertalot & Rumrich, 2000.

Witkowskia inexplorata (Krasske) Kulikovskiy, Glushchenko, Mironov & Kociolek comb. nov.

Basionym: Navicula inexplorata Krasske, 1939. The diatom flora of southern Chile. The Archive of Hydrobiology and Plankton studies, Stuttgart 35(3), p. 392; pl. 12, fig. 6.

Synonym: Placoneis inexplorata (Krasske) Lange-Bertalot in Metzeltin, Lange-Bertalot & García-Rodriguez, 2005.

Witkowskia insignita (Hustedt) Kulikovskiy, Glushchenko, Mironov & Kociolek comb. nov.

Basionym: Navicula insignita Hustedt, 1942b. Freshwater diatoms of the Indomalayan archipelago and the Hawaiian islands. According to the material of the Wallacea expedition. International Review of the entire Hydrobiology and Hydrography 42(1/3), p. 73, fig. 126, 139–141.

Synonym: Placoneis insignita (Hustedt) Cox, 2003.

Witkowskia insularis Sherbakova & Pomazkina in Pomazkina, Rodionova & Sherbakova Kulikovskiy, Glushchenko, Mironov & Kociolek comb. nov.

Basionym: Placoneis insularis Sherbakova & Pomazkina in Pomazkina, Rodionova & Sherbakova, 2019. Validation of 123 names of new diatom taxa from Lake Baikal. Limnology and Freshwater Biology 2019(1), p. 195.

Synonym: Placoneis insularis Sherbakova & Pomazkina in Pomazkina, Rodionova & Sherbakova, 2019, nom. invalid.

Witkowskia interglacialis (Hustedt) Kulikovskiy, Glushchenko, Mironov & Kociolek comb. nov.

Basionym: Navicula interglacialis Hustedt, 1944. New and little known Diatoms. Report of the German Botanical Society 61, p. 286, pl. VIII (Kezlya et al., 2022), fig. 27.

Synonym: Placoneis interglacialis (Hustedt) Cox, 2003.

Witkowskia ivanii (Pomazkina & Rodionova in Pomazkina, Rodionova & Sherbakova) Kulikovskiy, Glushchenko, Mironov & Kociolek comb. nov.

Basionym: Placoneis ivanii Pomazkina & Rodionova in Pomazkina, Rodionova & Sherbakova, 2019. Validation of 123 names of new diatom taxa from Lake Baikal. Limnology and Freshwater Biology 2019(1), p. 195.

Synonym: Placoneis ivanii Pomazkina & Rodionova in Pomazkina, Rodionova & Sherbakova, 2019, nom. invalid.

Witkowskia itamoemae (Straube, Tremarin & Ludwig in Straube, Tremarin, de Castro-Pires, Marquardt & Ludwig) Kulikovskiy, Glushchenko, Mironov & Kociolek comb. nov.

Basionym: Placoneis itamoemae Straube, Tremarin & Ludwig in Straube et al., 2013. Morphology, ultrastructure and distribution of Placoneis itamoemae sp. nov. (Cymbellaceae) from Brazil. Phytotaxa 76(3), p. 56, figs. 1–24.

Witkowskia izhboldinae (Vishnyakov in Vishnyakov, Kulikovskiy, Dorofeyuk & Genkal) Kulikovskiy, Glushchenko, Mironov & Kociolek comb. nov.

Basionym: Placoneis izhboldinae Vishnyakov in Vishnyakov et al., 2016. Botanicheskii Zhurnal, 1301, pl. I, figs. 1–8, pl. II, figs. 1–6.

Witkowskia jatobensis (Krasske) Kulikovskiy, Glushchenko, Mironov & Kociolek comb. nov.

Basionym: Navicula jatobensis Krasske, 1951. The diatom flora of the Açudas Northeastern Brazil (On the Biatom flora of Brazil II). Archives of Hydrobiology 44, p. 651; fig. 12.

Synonym: Placoneis jatobensis (Krasske) Metzeltin & Lange-Bertalot, 1998.

Witkowskia juriljii (Miho & Lange-Bertalot) Kulikovskiy, Glushchenko, Mironov & Kociolek comb. nov.

Basionym: Placoneis juriljii Miho & Lange-Bertalot, 2006. Diversity of the genus Placoneis in Lake Ohrid and other freshwater habitats of Albania. Proc. Int. Diatom. Symp. 18, p. 306, figs. 27–35.

Witkowskia lata (Peragallo in Tempère & Peragallo) Kulikovskiy, Glushchenko, Mironov & Kociolek comb. et stat nov.

Basionym: Navicula dicephala var. lata Peragallo in Tempère & Peragallo, 1908. Diatoms from All Over The World, 2nd Edition, 30 figures, fig. 2–7, p. 56; No. 103, 104.

Synonym: Placoneis lata (M. Peragallo) Lowe in Johansen et al., 2004.

Witkowskia laticuneata (Kulikovskiy, Lange-Bertalot & Metzeltin in Kulikovskiy, Lange-Bertalot, Metzeltin & Witkowski) Kulikovskiy, Glushchenko, Mironov & Kociolek comb. nov.

Basionym: Placoneis laticuneata Kulikovskiy, Lange-Bertalot & Metzeltin in Kulikovskiy et al., 2012. Lake Baikal: Hotspot of endemic diatoms I. Iconographia Diatomologica 23, p. 227, pl. 125, figs. 1–3.

Witkowskia latiuscula (Grunow in Cleve & Grunow) Kulikovskiy, Glushchenko, Mironov & Kociolek comb. nov. et stat. nov.

Basionym: Navicula gastrum var. latiusculum “latiuscul” Grunow in Cleve & Grunow, 1880. Contributions to the knowledge of arctic diatoms. Swedish Royal Society, Proceedings of the Academy of Sciences 17(2), p. 31.

Synonym: Placoneis latiuscula (Grunow) Kulikovskiy & Genkal in Kulikovskiy, Genkal & Mikheeva, 2010.

Witkowskia likhoshwayae (Metzeltin, Kulikovskiy & Lange-Bertalot in Kulikovskiy, Lange-Bertalot, Metzeltin & Witkowski) Kulikovskiy, Glushchenko, Mironov & Kociolek comb. nov.

Basionym: Placoneis likhoshwayae Metzeltin, Kulikovskiy & Lange-Bertalot in Kulikovskiy et al., 2012. Lake Baikal: Hotspot of endemic diatoms I. Iconographia Diatomologica 23, p. 227, pl. 132, fig. 1.

Witkowskia linearis (Pomazkina & Sherbakova in Pomazkina, Rodionova & Sherbakova) Kulikovskiy, Glushchenko, Mironov & Kociolek comb. nov.

Basionym: Placoneis linearis Pomazkina & Sherbakova in Pomazkina, Rodionova & Sherbakova, 2019. Validation of 123 names of new diatom taxa from Lake Baikal. Limnology and Freshwater Biology 2019(1), p. 195.

Synonym: Placoneis linearis Pomazkina & Sherbakova in Pomazkina, Rodionova & Sherbakova, 2018, nom. invalid.

Witkowskia lucinensis (Lange-Bertalot in Hofmann, Werum & Lange-Bertalot) Kulikovskiy, Glushchenko, Mironov & Kociolek comb. nov.

Basionym: Placoneis lucinensis Lange-Bertalot in Hofmann, Werum & Lange-Bertalot, 2011. Diatomeen im Süsswasser-Benthos von Mitteleuropa. Bestimmungsflora Kieselalgen für die ökologische Praxis. Über 700 der häufigsten Arten und ihre Ökologie, p. 501, pl. 47, figs. 15–19.

Witkowskia ludmilae (Pomazkina & Sherbakova in Pomazkina, Rodionova & Sherbakova) Kulikovskiy, Glushchenko, Mironov & Kociolek comb. nov.

Basionym: Placoneis ludmilae Pomazkina & Sherbakova in Pomazkina, Rodionova & Sherbakova, 2019. Validation of 123 names of new diatom taxa from Lake Baikal. Limnology and Freshwater Biology 2019(1), p. 195.

Synonym: Placoneis ludmilae Pomazkina & Sherbakova in Pomazkina, Rodionova & Sherbakova, 2018, nom. invalid.

Witkowskia macedonica (Levkov & Metzeltin in Levkov, Krstic, Metzeltin & Nakov) Kulikovskiy, Glushchenko, Mironov & Kociolek comb. nov.

Basionym: Placoneis macedonica Levkov & Metzeltin in Levkov et al., 2007. Diatoms of Lakes Prespa and Ohrid, about 500 taxa from ancient lake system. Iconographia Diatomologica 16, p.110, pl. 98; pl. 99, figs. 20–23.

Witkowskia maculata (Hustedt) Kulikovskiy, Glushchenko, Mironov & Kociolek comb. et stat. nov.

Basionym: Navicula placentula var. maculata Hustedt, 1945. Diatoms from lakes and headwaters of the Balkan Peninsula. Archives of Hydrobiology 40(4), p. 928, pl. XL, fig. 16.

Synonym: Placoneis maculata (Hustedt) Levkov in Levkov et al., 2007.

Witkowskia madagascariensis (Lange-Bertalot & Metzeltin in Metzeltin & Lange-Bertalot) Kulikovskiy, Glushchenko, Mironov & Kociolek comb. nov.

Basionym: Placoneis madagascariensis Lange-Bertalot & Metzeltin in Metzeltin & Lange-Bertalot, 2002. Diatoms from the “Island Continent” Madagascar. Iconographia Diatomologica 11, p. 54, pl. 27, figs. 37–40; pl. 28, fig. 4.

Witkowskia magna (Pomazkina & Sherbakova in Pomazkina, Rodionova & Sherbakova) Kulikovskiy, Glushchenko, Mironov & Kociolek comb. nov.

Basionym: Placoneis magna Pomazkina & Sherbakova in Pomazkina, Rodionova & Sherbakova, 2019. Validation of 123 names of new diatom taxa from Lake Baikal. Limnology and Freshwater Biology 2019(1), p. 195.

Synonym: Placoneis magna Pomazkina & Sherbakova in Pomazkina, Rodionova & Sherbakova, 2019, nom. invalid.

Witkowskia maior (Kulikovskiy, Lange-Bertalot & Metzeltin in Kulikovskiy, Lange-Bertalot, Metzeltin & Witkowski) Kulikovskiy, Glushchenko, Mironov & Kociolek comb. nov.

Basionym: Placoneis maior Kulikovskiy, Lange-Bertalot & Metzeltin in Kulikovskiy et al., 2012. Lake Baikal: Hotspot of endemic diatoms I. Iconographia Diatomologica 23, p. 228, pl. 124, figs. 4–7.

Witkowskia margaritae (Kulikovskiy & Lange-Bertalot in Kulikovskiy, Lange-Bertalot, Metzeltin & Witkowski) Kulikovskiy, Glushchenko, Mironov & Kociolek comb. nov.

Basionym: Placoneis margaritae Kulikovskiy & Lange-Bertalot in Kulikovskiy et al., 2012. Lake Baikal: Hotspot of endemic diatoms I. Iconographia Diatomologica 23, p. 229, pl. 128, figs. 1–14; pl. 129, figs. 1, 2; pl. 130, figs. 1, 2.

Witkowskia merinensis (Metzeltin, Lange-Bertalot & García-Rodríguez) Kulikovskiy, Glushchenko, Mironov & Kociolek comb. nov.

Basionym: Placoneis merinensis Metzeltin, Lange-Bertalot & García-Rodriguez, 2005. Diatoms of Uruguay. Compared with other taxa from South America and elsewhere. Iconographia Diatomologica 15, p. 183–184, pl. 72, figs. 7–11.

Witkowskia mira (Pomazkina & Rodionova in Pomazkina, Rodionova & Sherbakova) Kulikovskiy, Glushchenko, Mironov & Kociolek comb. nov.

Basionym: Placoneis mira Pomazkina & Rodionova in Pomazkina, Rodionova & Sherbakova, 2019. Validation of 123 names of new diatom taxa from Lake Baikal. Limnology and Freshwater Biology 2019(1), p. 196.

Synonym: Placoneis mira Pomazkina & Rodionova in Pomazkina, Rodionova & Sherbakova, 2018, nom. invalid.

Witkowskia miseranda (Kulikovskiy & Lange-Bertalot in Kulikovskiy, Lange-Bertalot, Metzeltin & Witkowski) Kulikovskiy, Glushchenko, Mironov & Kociolek comb. nov.

Basionym: Placoneis miseranda Kulikovskiy & Lange-Bertalot in Kulikovskiy et al., 2012. Lake Baikal: Hotspot of endemic diatoms I. Iconographia Diatomologica 23, p. 230, pl. 125, figs. 10, 11; pl. 132, figs. 13–15.

Witkowskia molesta (Metzeltin & Lange-Bertalot) Kulikovskiy, Glushchenko, Mironov & Kociolek comb. nov.

Basionym: Placoneis molesta Metzeltin & Lange-Bertalot, 1998. Tropical diatoms of South America I: About 700 predominantly rarely known or new taxa representative of the neotropical flora. Iconographia Diatomologica 5, p. 197–198, pl. 89, figs. 6–9.

Witkowskia molestissima (Metzeltin, Lange-Bertalot & García-Rodríguez) Kulikovskiy, Glushchenko, Mironov & Kociolek comb. nov.

Basionym: Placoneis molestissima Metzeltin, Lange-Bertalot & García-Rodriguez, 2005. Diatoms of Uruguay. Compared with other taxa from South America and elsewhere. Iconographia Diatomologica 15, p. 186, pl. 70, figs. 14–21; pl. 77, fig. 4, 4A.

Witkowskia mollis (Pomazkina & Rodionova in Pomazkina, Rodionova & Sherbakova, 2019) Kulikovskiy, Glushchenko, Mironov & Kociolek comb. nov.

Basionym: Placoneis mollis Pomazkina & Rodionova in Pomazkina, Rodionova & Sherbakova, 2019. Validation of 123 names of new diatom taxa from Lake Baikal. Limnology and Freshwater Biology 2019(1), p. 196.

Synonym: Placoneis mollis Pomazkina & Rodionova in Pomazkina, Rodionova & Sherbakova, 2019, nom. invalid.

Witkowskia nanoclementis (Lange-Bertalot & Wojtal) Kulikovskiy, Glushchenko, Mironov & Kociolek comb. nov.

Basionym: Placoneis nanoclementis Lange-Bertalot & Wojtal, 2014. Diversity in species complexes of Placoneis clementis (Grunow) Cox and Paraplaconeis placentula (Ehrenberg) Kulikovskiy, Lange-Bertalot & Metzeltin. Beihefte zur Nova Hedwigia 143, p. 407, figs. 14–28, 71–73.

Witkowskia navicula (Pomazkina & Sherbakova in Pomazkina, Rodionova & Sherbakova) Kulikovskiy, Glushchenko, Mironov & Kociolek comb. nov.

Basionym: Placoneis navicula Pomazkina & Sherbakova in Pomazkina, Rodionova & Sherbakova, 2019. Validation of 123 names of new diatom taxa from Lake Baikal. Limnology and Freshwater Biology 2019(1), p. 196.

Synonym: Placoneis navicula Pomazkina & Sherbakova in Pomazkina, Rodionova & Sherbakova, 2018, nom. invalid.

Witkowskia neglecta (R.M. Patrick) Kulikovskiy, Glushchenko, Mironov & Kociolek comb. et stat. nov.

Basionym: Navicula elginensis var. neglecta R.M. Patrick in Patrick & Reimer, 1966. The diatoms of the United States exclusive of Alaska and Hawaii. Volume 1: Fragilariaceae, Eunotiaceae, Achnanthaceae, Naviculaceae. Monographs of the Academy of Natural Sciences of Philadelphia 13: p. 525; pl. 50, fig. 5.

Synonym: Placoneis elginensis var. neglecta H. Kobayasi in Mayama et al., 2002.

Witkowskia neoexigua (Miho & Lange-Bertalot) Kulikovskiy, Glushchenko, Mironov & Kociolek comb. nov.

Basionym: Placoneis neoexigua Miho & Lange-Bertalot, 2006. Diversity of the genus Placoneis in Lake Ohrid and other freshwater habitats of Albania. Proc. Int. Diatom. Symp. 18, p. 302, 304, figs. 1–11, 20–26.

Witkowskia neotropica (Metzeltin & Lange-Bertalot) Kulikovskiy, Glushchenko, Mironov & Kociolek comb. nov.

Basionym: Placoneis neotropica Metzeltin & Lange-Bertalot, 1998. Tropical diatoms of South America I: About 700 predominantly rarely known or new taxa representative of the neotropical flora. Iconographia Diatomologica 5, p. 198–199, pl. 89, figs. 1–5.

Witkowskia nipponica (Skvortzow) Kulikovskiy, Glushchenko, Mironov & Kociolek comb. et stat. nov.

Basionym: Navicula similis var. nipponica Skvortzow, 1936. Diatoms from Biwa Lake, Honshu Island, Nippon. Philippine Journal of Science 61(2), p. 276, fig. 3: 2.

Witkowskia obtuseprotracta (Hustedt) Kulikovskiy, Glushchenko, Mironov & Kociolek comb. nov.

Basionym: Navicula obtuseprotracta Hustedt, 1964. The Diatoms of Germany, Austria and Switzerland, including other countries of Europe, as well as the bordering marine areas. In: L. Rabenhorst (ed.), Cryptogamous flora of Germany, Austria and Switzerland. Academic Publishers, Leipzig 7 (Teil 3, Lief. 4), p. 767, fig. 1740.

Synonym: Placoneis obtuseprotracta (Hustedt) Li & Metzeltin in Gong et al., 2013.

Witkowskia ohridana (Levkov & Metzeltin in Levkov, Krstic, Metzeltin & Nakov) Kulikovskiy, Glushchenko, Mironov & Kociolek comb. nov.

Basionym: Placoneis ohridana Levkov & Metzeltin in Levkov et al., 2007. Diatoms of Lakes Prespa and Ohrid, about 500 taxa from ancient lake system. Iconographia Diatomologica 16, p. 111, pl. 97, figs. 1–7.

Witkowskia opportuna (Hustedt) Kulikovskiy, Glushchenko, Mironov & Kociolek comb. nov.

Basionym: Navicula opportuna Hustedt, 1950. The Diatom flora of North German lakes with special consideration of the Holstein lake area V-VII. Lakes in Mecklenburg, Lauenburg and Northeast Germany. Archives of Hydrobiology 43, p. 436, pl. 39, figs. 21, 22.

Synonym: Placoneis opportuna (Hustedt) Chudaev & Gololobova, 2016.

Witkowskia ovillus (Metzeltin, Lange-Bertalot & García-Rodríguez) Kulikovskiy, Glushchenko, Mironov & Kociolek comb. nov.

Basionym: Placoneis ovillus Metzeltin, Lange-Bertalot & García-Rodriguez, 2005. Diatoms of Uruguay. Compared with other taxa from South America and elsewhere. Iconographia Diatomologica 15, p. 187–188, pl. 74, figs. 20–26.

Witkowskia paraelginensis (Lange-Bertalot in U. Rumrich, Lange-Bertalot & M. Rumrich) Kulikovskiy, Glushchenko, Mironov & Kociolek comb. nov.

Basionym: Placoneis paraelginensis Lange-Bertalot in Rumrich, Lange-Bertalot & Rumrich, 2000. Diatoms of the Andes. From Venezuela to Patagonia/Tierra del Fuego and two additional contributions. Iconographia Diatomologica 9, p. 208, pl. 60, figs. 17–20.

Witkowskia paragelegma (Pomazkina & Rodionova in Pomazkina, Rodionova & Sherbakova) Kulikovskiy, Glushchenko, Mironov & Kociolek comb. nov.

Basionym: Placoneis paragelegma Pomazkina & Rodionova in Pomazkina, Rodionova & Sherbakova, 2019. Validation of 123 names of new diatom taxa from Lake Baikal. Limnology and Freshwater Biology 2019(1), p. 196.

Synonym: Placoneis paragelegma Pomazkina & Rodionova in Pomazkina, Rodionova & Sherbakova, 2018, nom. invalid.

Witkowskia paraundulata (Hustedt) Kulikovskiy, Glushchenko, Mironov & Kociolek comb. et stat. nov.

Basionym: Navicula exigua f. undulata Hustedt, 1942b. Freshwater diatoms of the Indomalayan archipelago and the Hawaiian islands. International Review of the entire Hydrobiology and Hydrography 42(1/3), p. 73, fig. 135.

Synonym: Placoneis paraundulata Ohtsuka, 2002.

Witkowskia parazula (Kulikovskiy & Lange-Bertalot in Kulikovskiy, Lange-Bertalot, Metzeltin & Witkowski) Kulikovskiy, Glushchenko, Mironov & Kociolek comb. nov.

Basionym: Placoneis parazula Kulikovskiy & Lange-Bertalot in Kulikovskiy et al., 2012. Lake Baikal: Hotspot of endemic diatoms I. Iconographia Diatomologica 23, p. 231, pl. 131, figs. 14–17.

Witkowskia parunculus (Hustedt) Kulikovskiy, Glushchenko, Mironov & Kociolek comb. nov.

Basionym: Navicula parunculus Hustedt, 1952. Neue und wenig bekannte Diatomeen. IV. Botaniska Notiser, p. 398, fig. 115.

Synonym: Placoneis parunculus (Hustedt) Lange-Bertalot & Metzeltin in Metzeltin, Lange-Bertalot & García-Rodríguez, 2005.

Witkowskia parvapolonica (Lange-Bertalot & Wojtal) Kulikovskiy, Glushchenko, Mironov & Kociolek comb. nov.

Basionym: Placoneis parvapolonica Lange-Bertalot & Wojtal, 2014. Diversity in species complexes of Placoneis clementis (Grunow) Cox and Paraplaconeis placentula (Ehrenberg) Kulikovskiy, Lange-Bertalot & Metzeltin. Beihefte zur Nova Hedwigia 143, p. 408, figs. 29–44, 74–76.

Witkowskia parvula (Pomazkina & Rodionova in Pomazkina, Rodionova & Sherbakova) Kulikovskiy, Glushchenko, Mironov & Kociolek comb. nov.

Basionym: Placoneis parvula Pomazkina & Rodionova in Pomazkina, Rodionova & Sherbakova, 2019. Validation of 123 names of new diatom taxa from Lake Baikal. Limnology and Freshwater Biology 2019(1), p. 196.

Synonym: Placoneis parvula Pomazkina & Rodionova in Pomazkina, Rodionova & Sherbakova, 2018, nom. invalid.

Witkowskia patagonica (Maidana in Maidana, Aponte, Fey, Schäbitz & Morales) Kulikovskiy, Glushchenko, Mironov & Kociolek comb. nov.

Basionym: Placoneis patagonica Maidana in Maidana et al., 2017. Cyclostepanos salsae and Placoneis patagonica, two new diatoms (Bacillariophyta) from Laguna Cháltel in southern Patgonia, Argentina. Nova Hedwigia Beiheft 146, p. 98, figs. 21–40.

Witkowskia paucimarensis (Rodionova & Pomazkina in Pomazkina, Rodionova & Sherbakova) Kulikovskiy, Glushchenko, Mironov & Kociolek comb. nov.

Basionym: Placoneis paucimarensis Rodionova & Pomazkina in Pomazkina, Rodionova & Sherbakova, 2019. Validation of 123 names of new diatom taxa from Lake Baikal. Limnology and Freshwater Biology 2019(1), p. 196.

Synonym: Placoneis paucimarensis Rodionova & Pomazkina in Pomazkina, Rodionova & Sherbakova, 2018, nom. invalid.

Witkowskia pellaifa (Lange-Bertalot & U. Rumrich in U. Rumrich, Lange-Bertalot & M. Rumrich) Kulikovskiy, Glushchenko, Mironov & Kociolek comb. nov.

Basionym: Placoneis pellaifa Lange-Bertalot & U. Rumrich in Rumrich, Lange-Bertalot & Rumrich, 2000. Diatoms of the Andes. From Venezuela to Patagonia/Tierra del Fuego and two additional contributions. Iconographia Diatomologica 9, p. 210, pl. 60, figs. 9, 10.

Witkowskia perelginensis (Metzeltin, Lange-Bertalot & García-Rodríguez) Kulikovskiy, Glushchenko, Mironov & Kociolek comb. nov.

Basionym: Placoneis perelginensis Metzeltin, Lange-Bertalot & García-Rodriguez, 2005. Diatoms of Uruguay. Compared with other taxa from South America and elsewhere. Iconographia Diatomologica 15, p. 189, pl. 70, figs. 6–13.

Witkowskia poriphera (Hustedt) Kulikovskiy, Glushchenko, Mironov & Kociolek comb. nov.

Basionym: Navicula porifera Hustedt, 1944. New and little known Diatoms. Report of the German Botanical Society 61, p. 284, fig. 25.

Synonym: Placoneis porifera (Hustedt) Ohtsuka & Fujita, 2001.

Witkowskia potapovae (Kociolek) Kulikovskiy, Glushchenko, Mironov & Kociolek comb. nov.

Basionym: Placoneis potapovae Kociolek in Kociolek et al., 2014. Diatoms of the United States, 1 Taxonomy, ultrastructure and descriptions of new species and other rarely reported taxa from lake sediments in the western U.S.A. Bibliotheca Diatomologica 61: 23, pl. 28: figs. 14–20; pl. 30: figs. 3–4; pl. 31: figs. 1–4.

Witkowskia prespanensis (Levkov, Krstic, Nakov & Metzeltin in Levkov, Krstic, Metzeltin & Nakov) Kulikovskiy, Glushchenko, Mironov & Kociolek comb. nov.

Basionym: Placoneis prespanensis Levkov, Krstic Nakov & Metzeltin in Levkov et al., 2007. Diatoms of Lakes Prespa and Ohrid, about 500 taxa from ancient lake system. Iconographia Diatomologica 16, p.112, pl. 99; figs. 1–17, pl. 100.

Witkowskia pseudabundans (Levkov in Levkov & Williams) Kulikovskiy, Glushchenko, Mironov & Kociolek comb. nov.

Basionym: Placoneis pseudabundans Levkov in Levkov & Williams, 2011. Fifteen new diatom (Bacillariophyta) species from Lake Ohrid, Macedonia. Phytotaxa 30, p. 14, figs. 92–95.

Witkowskia producta (Pomazkina & Rodionova in Pomazkina, Rodionova & Sherbakova) Kulikovskiy, Glushchenko, Mironov & Kociolek comb. et stat. nov.

Basionym: Placoneis radialis var. producta Pomazkina & Rodionova in Pomazkina, Rodionova & Sherbakova, 2019. Validation of 123 names of new diatom taxa from Lake Baikal. Limnology and Freshwater Biology 2019(1), p. 196.

Synonym: Placoneis radialis var. producta Pomazkina & Rodionova in Pomazkina, Rodionova & Sherbakova, 2018, nom. invalid.

Witkowskia pseudoclementis (Hustedt) Kulikovskiy, Glushchenko, Mironov & Kociolek comb. nov.

Basionym: Navicula pseudoclementis Hustedt, 1952. New and little known Diatoms. IV. Botanical Notes, p. 400, fig. 107.

Synonym: Placoneis pseudoclementis (Hustedt) Lange-Bertalot in Metzeltin, Lange-Bertalot & García-Rodriguez, 2005.

Witkowskia pseudolacostris (Skabitschewsky) Kulikovskiy, Glushchenko, Mironov & Kociolek comb. nov.

Basionym: Navicula pseudolacustris Skabichevsky, 1936. New and interesting diatoms from the northern Baikal Lake. Botanical Journal 21(6), p. 713, 721, pl. 2, fig. 19.

Synonym: Placoneis pseudolacustris (Skabitschewsky) Kulikovskiy, Lange-Bertalot & Khursevich, 2014b.

Witkowskia pseudoporifera (Hustedt) Kulikovskiy, Glushchenko, Mironov & Kociolek comb. nov.

Basionym: Navicula pseudoporifera Hustedt, 1952. New and little known Diatoms. IV. Botanical Notes, p. 406, fig. 117.

Synonym: Placoneis pseudoporifera (Hustedt) Lange-Bertalot in Metzeltin, Lange-Bertalot & García-Rodriguez, 2005.

Witkowskia radialis (Pomazkina & Rodionova in Pomazkina, Rodionova & Sherbakova) Kulikovskiy, Glushchenko, Mironov & Kociolek comb. nov.

Basionym: Placoneis radialis Pomazkina & Rodionova in Pomazkina, Rodionova & Sherbakova, 2019. Validation of 123 names of new diatom taxa from Lake Baikal. Limnology and Freshwater Biology 2019(1), p. 196.

Synonym: Placoneis radialis Pomazkina & Rodionova in Pomazkina, Rodionova & Sherbakova, 2018, nom. invalid.

Witkowskia regionalis (Pomazkina & Sherbakova in Pomazkina, Rodionova & Sherbakova) Kulikovskiy, Glushchenko, Mironov & Kociolek comb. nov.

Basionym: Placoneis regionalis Pomazkina & Sherbakova in Pomazkina, Rodionova & Sherbakova, 2019. Validation of 123 names of new diatom taxa from Lake Baikal. Limnology and Freshwater Biology 2019(1), p. 197.

Synonym: Placoneis regionalis Pomazkina & Sherbakova in Pomazkina, Rodionova & Sherbakova, 2018, nom. invalid.

Witkowskia reimeri (Kociolek & Thomas) Kulikovskiy, Glushchenko, Mironov & Kociolek comb. nov.

Basionym: Placoneis reimeri Kociolek & Thomas, 2010. Taxonomy and ultrastructure of five naviculoid diatoms (class Bacillariophyceae) from the Rocky Mountains of Colorado (USA), with the description of a new genus and four new species. Nova Hedwigia 90(1/2), p. 201, figs. 30–35, 39–41.

Witkowskia rhombea Pomazkina & Rodionova in Pomazkina, Rodionova & Sherbakova Kulikovskiy, Glushchenko, Mironov & Kociolek comb. nov.

Basionym: Placoneis rhombea Pomazkina & Rodionova in Pomazkina, Rodionova & Sherbakova, 2019. Validation of 123 names of new diatom taxa from Lake Baikal. Limnology and Freshwater Biology 2019(1), p. 197.

Synonym: Placoneis rhombea Pomazkina & Rodionova in Pomazkina, Rodionova & Sherbakova, 2019, nom. invalid.

Witkowskia rhombelliptica (Metzeltin, Lange-Bertalot & García-Rodríguez) Kulikovskiy, Glushchenko, Mironov & Kociolek comb. nov.

Basionym: Placoneis rhombelliptica Metzeltin, Lange-Bertalot & García-Rodriguez, 2005. Diatoms of Uruguay. Compared with other taxa from South America and elsewhere. Iconographia Diatomologica 15, p. 193–194, pl. 71, figs. 16–23, pl. 76, fig. 2.

Witkowskia rostrata (A. Mayer) Kulikovskiy, Glushchenko, Mironov & Kociolek comb. et stat. nov.

Basionym: Navicula dicephala var. rostrata Mayer, 1917. Contributions to the diatom flora of Bavaria. Part II, Bacillariales by Dillingen A. Donau. Memoirs of the Royal Bavarian Botanical Society in Regensburg 13, p. 114, pl. 1, fig. 42a, 42b.

Synonyms: Placoneis rostrata (A. Mayer) Cox, 2003.

Witkowskia ruppeliana (Metzeltin, Kulikovskiy & Lange-Bertalot in Kulikovskiy, Lange-Bertalot, Metzeltin & Witkowski) Kulikovskiy, Glushchenko, Mironov & Kociolek comb. nov.

Basionym: Placoneis ruppeliana Metzeltin, Kulikovskiy & Lange-Bertalot in Kulikovskiy et al., 2012. Lake Baikal: Hotspot of endemic diatoms I. Iconographia Diatomologica 23, 232, pl. 125: figs. 4–9; pl. 126: figs. 1, 2; pl. 127: figs. 1–3.

Witkowskia santaremensis (Metzeltin & Lange-Bertalot) Kulikovskiy, Glushchenko, Mironov & Kociolek comb. nov.

Basionym: Placoneis santaremensis Metzeltin & Lange-Bertalot, 1998. Tropical diatoms of South America I: About 700 predominantly rarely known or new taxa representative of the neotropical flora. Iconographia Diatomologica 5, 199–200; pl. 75, figs. 18–24.

Witkowskia scharfii (Lange-Bertalot & U. Rumrich in U. Rumrich, Lange-Bertalot and M.U. Rumrich) Kulikovskiy, Glushchenko, Mironov & Kociolek comb. nov.

Basionym: Placoneis scharfii Lange-Bertalot & U. Rumrich in Rumrich, Lange-Bertalot & Rumrich, 2000. Diatoms of the Andes. From Venezuela to Patagonia/Tierra del Fuego and two additional contributions. Iconographia Diatomologica 9, p. 211; pl. 59, figs. 5–7.

Witkowskia septentrionalis (Pomazkina & Rodionova in Pomazkina, Rodionova & Sherbakova) Kulikovskiy, Glushchenko, Mironov & Kociolek comb. nov.

Basionym: Placoneis septentrionalis Pomazkina & Rodionova in Pomazkina, Rodionova & Sherbakova, 2019. Validation of 123 names of new diatom taxa from Lake Baikal. Limnology and Freshwater Biology 2019(1), p. 197.

Synonym: Placoneis septentrionalis Pomazkina & Rodionova in Pomazkina, Rodionova & Sherbakova, 2018, nom. invalid.

Witkowskia serena (Frenguelli) Kulikovskiy, Glushchenko, Mironov & Kociolek comb. nov.

Basionym: Navicula serena Frenguelli, 1941. Diatoms of the River Plate. Magazine of the Museum of River Plate, New Series, Botanical Section 3, p. 255, pl. 2, figs. 1–5.

Synonym: Placoneis serena (Frenguelli) Metzeltin in Metzeltin, Lange-Bertalot & García-Rodriguez, 2005.

Witkowskia signata (Hustedt) Kulikovskiy, Glushchenko, Mironov & Kociolek comb. et stat. nov.

Basionym: Navicula anglica var. signata Hustedt, 1944. New and little known Diatoms. Report of the German Botanical Society 61, p. 287, fig. 26.

Synonyms: Navicula pseudanglica var. signata (Hustedt) Lange-Bertalot Krammer & Lange-Bertalot, 1985; Placoneis anglophila var. signata (Hustedt) Lange-Bertalot in Metzeltin, Lange-Bertalot & García-Rodriguez, 2005.

Witkowskia signatoides (Metzeltin and Levkov in Levkov, Krstic, Metzeltin & Nakov) Kulikovskiy, Glushchenko, Mironov & Kociolek comb. nov.

Basionym: Placoneis signatoides Metzeltin and Levkov in Levkov et al., 2007. Diatoms of Lakes Prespa and Ohrid, about 500 taxa from ancient lake system. Iconographia Diatomologica 16, 114; pl. 90, figs. 2–9.

Witkowskia simplex (Pomazkina & Rodionova in Pomazkina, Rodionova & Sherbakova) Kulikovskiy, Glushchenko, Mironov & Kociolek comb. nov.

Basionym: Placoneis simplex Pomazkina & Rodionova in Pomazkina, Rodionova & Sherbakova, 2019. Validation of 123 names of new diatom taxa from Lake Baikal. Limnology and Freshwater Biology 2019(1), p. 197.

Synonym: Placoneis simplex Pomazkina & Rodionova in Pomazkina, Rodionova & Sherbakova, 2018, nom. invalid.

Witkowskia sinensis (Li & Metzeltin in Gong, Li, Metzeltin & Lange-Bertalot) Kulikovskiy, Glushchenko, Mironov & Kociolek comb. nov.

Basionym: Placoneis sinensis Li & Metzeltin in Gong et al., 2013. New species of Cymbella and Placoneis (Bacillariophyta) from late Pleistocene fossil, China. Phytotaxa 150(1), p. 34, figs. 40–43.

Witkowskia solaris (Pomazkina, Rodionova & Sherbakova) Kulikovskiy, Glushchenko, Mironov & Kociolek comb. nov.

Basionym: Placoneis solaris Pomazkina, Rodionova & Sherbakova, 2019. Validation of 123 names of new diatom taxa from Lake Baikal. Limnology and Freshwater Biology 2019(1): p. 197.

Witkowskia sovereigniae (Hustedt) Kulikovskiy, Glushchenko, Mironov & Kociolek comb. nov.

Basionym: Navicula sovereigniae Hustedt, 1955b. Marine littoral diatoms of Beaufort, North Carolina. Bulletin Duke University Marine Station 6: p. 25, pl. 8: figs. 18–20.

Synonym: Placoneis sovereignae (Hustedt) Torgan, Donadel & Gonçalves da Silva, 2010.

Witkowskia spinosa (Metzeltin, Kulikovskiy & Lange-Bertalot in Kulikovskiy, Lange-Bertalot, Metzeltin & Witkowski) Kulikovskiy, Glushchenko, Mironov & Kociolek comb. nov.

Basionym: Placoneis spinosa Metzeltin, Kulikovskiy & Lange-Bertalot in Kulikovskiy et al., 2012. Lake Baikal: Hotspot of endemic diatoms I. Iconographia Diatomologica 23, p. 233, pl. 122: figs. 1–4; pl. 123: figs. 1, 2.

Witkowskia subcapitata (Grunow) Kulikovskiy, Glushchenko, Mironov & Kociolek comb. et stat. nov.

Basionym: Navicula dicephala var. subcapitata Grunow, 1882. Contributions to the knowledge of fossil diatoms of Austria-Hungary. In: Contributions to Paleontology of Austria-Hungary and the East. II Band Pt 4. (Mojsisovics, E. & Neumayr, N. Eds), p. 156, pl. 30: fig. 54.

Synonym: Placoneis dicephala var. subcapitata (Grunow) Mereschkowsky, 1903.

Witkowskia subclementis (Hustedt) Kulikovskiy, Glushchenko, Mironov & Kociolek comb. nov.

Basionym: Navicula subclementis Hustedt, 1952. New and little known Diatoms. IV. Botanical Notes, p. 400; fig. 108.

Synonym: Placoneis subclementis (Hustedt) Lange-Bertalot in Metzeltin, Lange-Bertalot & García-Rodriguez, 2005.

Witkowskia subelegans (Levkov) Kulikovskiy, Glushchenko, Mironov & Kociolek comb. nov.

Basionym: Placoneis subelegans Levkov in Levkov & Williams, 2011. Fifteen new diatom (Bacillariophyta) species from Lake Ohrid, Macedonia. Phytotaxa 30: p. 16, figs. 97–113.

Witkowskia subgastriformis (Hustedt) Kulikovskiy, Glushchenko, Mironov & Kociolek comb. nov.

Basionym: Navicula subgastriformis Hustedt, 1945. Diatoms from lakes and headwaters of the Balkan Peninsula. Archives of Hydrobiology 40(4), p. 928, pl. XLII (42), figs. 13, 14.

Synonym: Placoneis subgastriformis (Hustedt) Cox, 2003.

Witkowskia subtilis (Kulikovskiy & Lange-Bertalot in Kulikovskiy, Lange-Bertalot, Metzeltin & Witkowski) Kulikovskiy, Glushchenko, Mironov & Kociolek comb. nov.

Basionym: Placoneis subtilis Kulikovskiy & Lange-Bertalot in Kulikovskiy et al., 2012. Lake Baikal: Hotspot of endemic diatoms I. Iconographia Diatomologica 23, p. 234, pl. 131: figs. 23–28; pl. 134: fig. 6.

Witkowskia subundulata (Kezlya, Glushchenko, Kulikovskiy & Kociolek) Kulikovskiy, Glushchenko, Mironov & Kociolek comb. nov.

Basionym: Placoneis subundulata Kezlya, Glushchenko, Kulikovskiy & Kociolek in Kezlya et al., 2021. Three New Species of Placoneis Mereschkowsky (Bacillariophyceae: Cymbellales) with Comments on Cryptic Diversity in the P. elginensis—Group. Water 2021, 13, p. 8, figs. 6–7.

Witkowskia surinamensis (P.T. Cleve) Kulikovskiy, Glushchenko, Mironov & Kociolek comb. nov.

Basionym: Navicula surinamensis Cleve, 1895. Synopsis of the Naviculoid Diatoms, Part II. Kongliga Svenska-Vetenskaps Akademiens Handlingar 27(3), p. 9; pl. 2, fig. 1.

Synonym: Placoneis surinamensis (P.T. Cleve) Metzeltin & Lange-Bertalot, 1998.

Witkowskia symmetrica (Hustedt) Kulikovskiy, Glushchenko, Mironov & Kociolek comb. et stat. nov.

Basionym: Navicula constans var. symmetrica Hustedt, 1957. Die Diatomeenflora des Fluß-systems der Weser im Gebiet der Hansestadt Bremen. Abhandlungen der Naturwissenschaftlichen Verein zu Bremen 34(3), p. 289, figs. 40, 41.

Synonyms: Placoneis constans var. symmetrica (Hustedt) Kobayasi in Mayama et al., 2002; Placoneis symmetrica (Hustedt) Lange-Bertalot in Metzeltin, Lange-Bertalot & García-Rodriguez, 2005.

Witkowskia tersa (Hustedt) Kulikovskiy, Glushchenko, Mironov & Kociolek comb. nov.

Basionym: Navicula tersa Hustedt, 1956. Diatoms from Lake Maracaibo in Venezuela. In: Results of the German limnological Venezuela expedition 1952 (F. Gessner & V. Vareschi). German Science Publishment, Berlin 1, p. 166, figs. 38–39.

Synonym: Placoneis tersa (Hustedt) Metzeltin & Lange-Bertalot, 1998.

Witkowskia tsendeekhuui (Metzeltin, Lange-Bertalot & Nergui) Kulikovskiy, Glushchenko, Mironov & Kociolek comb. nov.

Basionym: Placoneis tsendeekhuui Metzeltin, Lange-Bertalot & Soninkhishig, 2009. Diatoms in Mongolia. Iconographia Diatomologica 20, p. 83, pl. 55, figs. 1–6.

Witkowskia tumidula (Levkov in Levkov, Krstic, Metzeltin & Nakov) Kulikovskiy, Glushchenko, Mironov & Kociolek comb. nov.

Basionym: Placoneis tumidula Levkov in Levkov et al., 2007. Diatoms of Lakes Prespa and Ohrid, about 500 taxa from ancient lake system. Iconographia Diatomologica 16, p. 115; pl. 91, figs. 11–15; pl. 92, figs. 4–10; pl. 93, fig. 1.

Witkowskia undulata (Østrup) Kulikovskiy, Glushchenko, Mironov & Kociolek comb. et stat. nov.

Basionym: Navicula dicephala var. undulata Østrup, 1918. Fresh-water Diatoms from Iceland. In: The Botany of Iceland, vol. II, part I. (Rosenvinge, L.K. & Warming, E. Eds), p. 25, pl. 3, fig. 33.

Synonyms: Placoneis undulata Lange-Bertalot & U. Rumrich in Rumrich, Lange-Bertalot & Rumrich, 2000.

Witkowskia uruguayensis (Metzeltin, Lange-Bertalot & García-Rodríguez) Kulikovskiy, Glushchenko, Mironov & Kociolek comb. nov.

Basionym: Placoneis uruguayensis Metzeltin, Lange-Bertalot & García-Rodriguez, 2005. Diatoms of Uruguay. Compared with other taxa from South America and elsewhere. Iconographia Diatomologica 15, p. 197; pl. 78, figs. 1–4.

Witkowskia vadosa (Pomazkina & Rodionova in Pomazkina, Rodionova & Sherbakova) Kulikovskiy, Glushchenko, Mironov & Kociolek comb. nov.

Basionym: Placoneis vadosa Pomazkina & Rodionova in Pomazkina, Rodionova & Sherbakova, 2019. Validation of 123 names of new diatom taxa from Lake Baikal. Limnology and Freshwater Biology 2019(1), p. 197.

Synonym: Placoneis vadosa Pomazkina & Rodionova in Pomazkina, Rodionova & Sherbakova, 2018, nom. invalid.

Witkowskia vicina (Hustedt) Kulikovskiy, Glushchenko, Mironov & Kociolek comb. nov.

Basionym: Navicula vicina Hustedt, 1952. Neue und wenig bekannte Diatomeen. IV. Botaniska Notiser, p. 407; fig. 128.

Synonym: Placoneis vicina (Hustedt) Metzeltin & Lange-Bertalot, 1998.

Witkowskia vladimiri (Pomazkina in Pomazkina, Rodionova & Sherbakova) Kulikovskiy, Glushchenko, Mironov & Kociolek comb. nov.

Basionym: Placoneis vladimiri Pomazkina in Pomazkina, Rodionova & Sherbakova, 2019. Validation of 123 names of new diatom taxa from Lake Baikal. Limnology and Freshwater Biology 2019(1), p. 197.

Synonym: Placoneis vladimiri Pomazkina in Pomazkina, Rodionova & Sherbakova, 2018, nom. invalid.

Witkowskia waernensis (Foged) Kulikovskiy, Glushchenko, Mironov & Kociolek comb. nov.

Basionym: Navicula waernensis Foged, 1964. Freshwater Diatoms from Spitsbergen. Tromsö Museums Skrifter, Universitetsførlaget, Tromsö/Oslo 11, p. 106, pl. 9, fig. 12.

Synonym: Placoneis waernensis (Foged) Lange-Bertalot & Metzeltin in Metzeltin, Lange-Bertalot & Soninkhishig, 2009.

Witkowskia witkowskii (Metzeltin, Lange-Bertalot & García-Rodríguez) Kulikovskiy, Glushchenko, Mironov & Kociolek comb. nov.

Basionym: Placoneis witkowskii Metzeltin, Lange-Bertalot & García-Rodriguez, 2005. Diatoms of Uruguay. Compared with other taxa from South America and elsewhere. Iconographia Diatomologica 15, p. 200, pl. 71, figs. 8–15.

Witkowskia yucatanensis (Lange-Bertalot in Metzeltin & Lange-Bertalot) Kulikovskiy, Glushchenko, Mironov & Kociolek comb. nov.

Basionym: Placoneis yucatanensis Lange-Bertalot in Metzeltin & Lange-Bertalot, 2007. Tropical Diatoms of South America II. Special remarks on biogeography disjunction. Iconographia Diatomologica 18, 233, pl. 114, figs. 7–11.

Witkowskia zimmermannii (Metzeltin & Lange-Bertalot) Kulikovskiy, Glushchenko, Mironov & Kociolek comb. nov.

Basionym: Placoneis zimmermannii Metzeltin & Lange-Bertalot, 1998. Tropical diatoms of South America I: About 700 predominantly rarely known or new taxa representative of the neotropical flora. Iconographia Diatomologica 5, p. 201–202; pl. 89, figs. 10–13.

Witkowskia zula (Kulikovskiy, Lange-Bertalot & Metzeltin in Kulikovskiy, Lange-Bertalot, Metzeltin & Witkowski) Kulikovskiy, Glushchenko, Mironov & Kociolek comb. nov.

Basionym: Placoneis zula Kulikovskiy, Lange-Bertalot & Metzeltin in Kulikovskiy et al., 2012. Lake Baikal: Hotspot of endemic diatoms I. Iconographia Diatomologica 23, p. 235, pl. 131: figs. 1–13.

Discussion

Diatom systematics is a rapidly changing field of science. It is well known that diatoms are difficult to culture, but the features of diatom frustule are constant. Thus, diatom systematics is based on the differences in valve morphology, and, primarily, differences in the structure of pore occlusions. The idea to separate genera of the order Cymbellales based on the latter feature has already been approached by several diatomists, e.g. Mann (1984) and Cox (2004). In the current article, based on comprehensive material, we have significantly upgraded the classification of pore occlusions, developed by Cox (2004), by using both morphological and molecular analyses.

Molecular tree, presented in this article (Fig. 1), includes numerous genera with unique features of the valve. These taxa were originally described based only on morphological data, in accordance to traditional perception of diatom systematics. At the same time, their statuses (as independent genera) are widely adopted. For example, Paraplaconeis, which was discovered by us earlier (Kulikovskiy et al., 2012), was proposed based solely on its unique morphology, more precisely, the structure of striae and areolae. The permanence of morphological features of the valve in this genus was later proved in several studies (Lange-Bertalot & Wojtal, 2014; Vishnyakov et al., 2016; Kezlya et al., 2021). Here, Paraplaconeis is investigated with the help of molecular analysis. Its results once again prove Paraplaconeis to be a separate, monophyletic genus. The situation is similar with other relatives of Placoneis. Studies by Cox (2003, 2004) and Jahn (2004) prove that features of the valve, including pore occlusions, are constant in the studied genera. Moreover, Cox (2004) justified the permanence of pore occlusions and described a new type of occlusion–tectulum–as specific for Placoneis. Therefore, we use a new classification of pore occlusions to revaluate Placoneis and describe new genera Chudaevia gen. nov. and Witkowskia gen. nov. It is worth mentioning that the newly described taxa can be separated not only by the differences in pore occlusions, but, as well, by other valve features, visible in LM and SEM. Besides, molecular analysis shows Witkowskia gen. nov. as a monophyletic group, independent from other genera of Cymbellales (Bayesian Post Priori = 0,98). Unfortunately, molecular data is currently unavailable for four species, that should remain in the genus Placoneis (Placoneis gastrum, Placoneis amphiboliformis, Placoneis coloradensis, and Placoneis elinae sp. nov.) and two species, transferred to Chudaevia gen. nov. (Chudaevia densistriata sp. nov., Chudaevia flabellata comb. nov.). At the same time, the listed species are morphologically different from other species of Placoneis. Because of that, introduction of new genera for them, even supported by morphology only, is just a matter of time, as demonstrated by Reichardt (2018).

The taxonomic instability of the genus Placoneis, which is discussed in this article, is associated with insufficient morphological investigation of pore occlusions in the type species of Placoneis and, in general, with the adoption of a broader interpretation of the pore occlusions structure following Cox (2004). Primarily, this problem can be explained by the wide representation of species of the genus Placoneis and lack of studies on newly described genera: Geissleria, Khursevichia Kulikovskiy, Lange-Bertalot & Metzeltin in Kulikovskiy et al., Ochigma Kulikovskiy, Lange-Bertalot & Metzeltin in Kulikovskiy et al., Rexlowea Kociolek and E.W. Thomas, Paraplaconeis, as well as Witkowskia gen. nov. and Chudaevia gen. nov., introduced here. In the second place, molecular investigations, carried out lately (Kulikovskiy et al., 2014a; Kezlya et al., 2020, 2021, 2022), indicate that recently discovered genera are independent, but phylogenetically close to each other, and that will be discussed below.

Different types of pore occlusions, mentioned in this article are listed in Table 4 with the genera, in which these types are present. The morphology of different pore occlusions is described below.

Table 4 Different types of pore occlusions in Placoneis and related genera.

Genus	Type species	Type of pore
occlusions	Characteristic of pore occlusions	
Witkowskia gen. nov.	Witkowskia neohambergii (Glushchenko, Kezlya, Kulikovskiy and Kociolek) Kulikovskiy, Glushchenko, Mironov and Kociolek comb. nov.	Tectulum	Several regularly arranged small struts, extending in the areolar opening perpendicular to the valve surface	
Chudaevia gen. nov.	Chudaevia densistriata Kulikovskiy, Mironov, Genkal, Glushchenko and Kociolek sp. nov.	Paratectulum	2–4 struts extending into the lumen of the openings parallel to the valve surface	
Paraplaconeis Kulikovskiy, Lange-Bertalot and Metzeltin in Kulikovskiy, Lange-Bertalot, Metzeltin and Witkowski	Paraplaconeis kornevae Kulikovskiy, Gusev and Lange-Bertalot in Kulikovskiy, Lange-Bertalot, Metzeltin and Witkowski	Foriculotectulum	A singular projection of irregular shape covering most of the areola, leaving a small opening	
Ochigma Kulikovskiy, Lange-Bertalot and Metzeltin in Kulikovskiy, Lange-Bertalot, Metzeltin and Witkowski	Ochigma baicalensis Kulikovskiy, Lange-Bertalot and Metzeltin in Kulikovskiy, Lange-Bertalot, Metzeltin and Witkowski	Oculus	Irregularly shaped papillary outgrowths, lying in a round shallow depression around the areola	
Khursevichia Kulikovskiy, Lange-Bertalot and Metzeltin in Kulikovskiy, Lange-Bertalot, Metzeltin and Witkowski	Khursevichia galinae Kulikovskiy, Lange-Bertalot and Metzeltin in Kulikovskiy, Lange-Bertalot, Metzeltin and Witkowski	Parvutectulum	2–3 struts lying in shallow circular depressions around the areola	
Geissleria Lange-Bertalot and Metzeltin	Geissleria moseri Metzeltin, Witkowski and Lange-Bertalot in Lange-Bertalot and Metzeltin	Annulus	4–10 silica outgrowths surrounding the large areolar opening (in the subpolar area of the valve)	
Placoneis Mereschkowsky	Placoneis gastrum (Ehrenberg) Mereschkowsky	Pseudotectulum	Several (usually >10) elongated struts of irregular shape and orientation surrounding the areolar opening	
Rexlowea Kociolek and E.W. Thomas	Rexlowea navicularis Kociolek and E.W. Thomas	Pseudovola	Small projections growing from the areolar walls, not branching	
Note:

The table contains new and previously described types of pore occlusions found in Placoneis and related genera.

A type of pore occlusion, originally described as tectulum by Cox (2004) is “an inner round or squarish flap-like covering, attached to the edges of an areola by several, regularly arranged, small struts”. Struts of tectulum are positioned perpendicular to the valve surface (Fig. 10A). This morphological structure is quite invariable and can be found in most species of Placoneis sensu stricto. However, this type of pore occlusions does not occur in the type species of Placoneis–P. gastrum. Instead, in P. gastrum pores are occluded by a structure (Fig. 10F) that can be described as a type of velum, formed by several (usually >10) elongated struts of irregular shape and orientation. Struts might be covered with thin flaps that break down during sample preparation. We propose a term “pseudotectulum” for this structure. In this case, tectulum, introduced by Cox (2004) is typical for diatoms from Witkowskia gen. nov. It should be noted that tectula also occur in the genus Geissleria, where tectula occlude areolae in striae (Fig. 10I) and isolated pores at the valve poles (Fig. 10J), but do not cover subpolar areolae (Fig. 10E).

Figure 10 (A–J) Pore occlusions in several genera of diatoms.

(A) Tectulum in Witkowskia gen. nov. (B) Foriculotectulum in Paraplaconeis. (C) Oculus in Ochigma. (D) Parvutectulum in Khursevichia. (E) Annulus in Geissleria (Kulikovskiy, Lange-Bertalot & Khursevich, 2014b). (F) Pseudotectulum in Placoneis. (G) Paratectulum in Chudaevia gen. nov. (H) Pseudovola in Rexlowea. (I) Areolae with tectula in Geissleria (Kulikovskiy, Lange-Bertalot & Khursevich, 2014b). (J) Isolated polar pores, occluded by tectula in Geissleria (Kulikovskiy, Lange-Bertalot & Khursevich, 2014b). Scale bars (A, B, D, F) 0.5 µm; (C, G, H) 0.25 µm; (E, I, J) 0.15 µm.

As the result, a lot of species, previously associated with Placoneis sensu lato, should be transferred to Witkowskia gen. nov. In this article, we present the new combinations for 168 species and varieties, previously regarded as members of the genus Placoneis. Witkowskia neohambergii (bas. Placoneis neohambergii Glushchenko, Kezlya, Kulikovskiy & Kociolek) comb. nov. is selected as type species of the new genus, because its morphology and structure of pore occlusions was studied in detail in Kezlya et al. (2021).

Hence, Placoneis sensu stricto currently includes only four species with similar morphology: Placoneis gastrum, Placoneis amphiboliformis, Placoneis coloradensis, and Placoneis elinae sp. nov.

Another taxon described in this article is Chudaevia densistriata sp. nov. The morphology of the new species is very similar to Placoneis flabellata. This taxon has been formerly regarded as the member of Navicula sensu lato until being placed in the genus Placoneis based upon features of chloroplast and raphe morphology (Kimura, Fukushima & Kobayashi, 2015). However, the organization of pore occlusions was poorly studied in that article. The newly described species was found in the samples from Ba Bể Lake (Bắc Kạn Province, Vietnam). During SEM investigation of this species, a peculiar type of pore occlusions was discovered. To include the former member of Placoneis, Placoneis flabellata, and the new species from Vietnam, a new genus, Chudaevia gen. nov., is introduced. The representatives of the new genus can be recognized by the presence of paratectulum (Fig. 10G). The paratectulum consists of 2–4 struts extending into the lumen of the opening, thus creating an illusion of S-shaped opening of areolae. Struts of the paratectulum lie parallel to the valve surface.

The genus Paraplaconeis, originally discovered in Lake Baikal (Kulikovskiy et al., 2012), has a relatively wide distribution in Eurasia, including the newly described species (Lange-Bertalot & Wojtal, 2014; Vishnyakov et al., 2016; Pomazkina, Rodionova & Sherbakova, 2019; Reichardt, 2021). Initially, this genus was separated from Placoneis sensu lato due to a different morphology of its pore occlusions. In Paraplaconeis, occlusions are formed by a singular projection of irregular shape (Fig. 10B). It covers most of the areolar opening but does not reach the opposite side of areola. We suggest giving a term foriculotectulum for this type of pore occlusion. The Foriculotectulum differs from a foriculum by lacking a narrow base of the projection. It is specific for pentagon-shaped areolae of Paraplaconeis.

The genus Ochigma was also described from Lake Baikal. Diatoms of this genus are characterized by very large sizes and coarse areolae. Comprehensive investigation of morphology and pore occlusions structure was carried out subsequently. Pore occlusions (Fig. 10C) in Ochigma are formed by irregularly shaped papillary outgrowths, that lie in a round shallow depression and surround the opening of areola. Interestingly, occlusions in Ochigma develop on the inner side of the valve. We propose naming this structure an “oculus”.

Another genus described from Lake Baikal is Khursevichia. This genus of small-sized valves is characterized by pore occlusions (Fig. 10D) that are relatively similar to tectula. However, each areola is formed by only 2–3 struts that lie in shallow circular depressions and surround a small opening of the areola. We propose the term parvutectulum for this type of pore occlusions.

The genus Rexlowea is a taxon with a few species known from Arctic zone (Kociolek & Thomas, 2010; Kociolek et al., 2018). This genus can be distinguished by a special type of pore occlusions that we propose naming “pseudovola”. Pseudovola (Fig. 10H) is a type of velum which is similar to a vola, but is formed by smaller projections, growing from the areolar walls. Also, projections of pseudovolae are less branched.

In comparison to previously described types of pore occlusions, that occlude all areolae in valves of genera listed above, the pore occlusions in Geissleria are more variable. Areolae of the genus Geissleria are very similar to areolae of dorsiventral cymbelloid diatoms, e.g., Encyonema. Particularly, in both taxa vimines are equipped with stubs and struts (Fig. 10I). At the same time, this genus was described based on the presence of annulus. The Annulus (Fig. 10E) should be treated as a structure, composed of subpolar areolae with large, elongated openings, possessing 4–10 silica outgrowths. Thus, the morphology of the annulus resembles a tectulum, however, the annulus is larger (usually discernible in LM) and can be found only in the subapical region of the valve. An annulus is typical for Geissleria. Additionally, valve poles in this genus are equipped with isolated pores, that are occluded by tectula (Fig. 10J) (Kulikovskiy et al., 2014a).

An important role of pore occlusions in understanding of diatom phylogeny was originally highlighted by Mann (1981). According to him, the structure of pore occlusions should be similar in monophyletic groups of diatoms. Cox (2004) recognized Mann’s ideas and introduced two new types of pore occlusions (vela), represented in several cymbelloid and naviculoid genera. It is important to mention that the term “velum” should be treated widely, so that cribrum, rota, vola, hymen, foriculum, etc. should be understood as variations of velum. Additional types of vela are suggested during our discussion above.

First phylogenetic studies of Placoneis suggested its close connections to cymbelloid diatoms (Bruder & Medlin, 2007; Kermarrec et al., 2011). Bruder & Medlin (2007) investigated the phylogeny of this genus and discussed the structure of pore occlusions, as well as morphological similarities between chloroplasts in Placoneis and cymbelloid taxa. Then, the phylogenetic relationships among cymbelloid diatoms was studied by Kulikovskiy et al. (2014a). It was demonstrated that Geissleria is strongly associated with cymbelloid diatoms, but its closest relative is Placoneis (Kulikovskiy et al., 2014a). Another related, however separate, clade is comprised of the species from the genus Paraplaconeis, studied in Kezlya et al. (2021).

These previous studies, and the phylogeny reported herein, support the importance of the pore occlusions as a morphological character, which can be used to diagnose monophyletic groups. The features listed in Table 4 appear as synapomorphies for each of the genera listed, some of which are currently included in molecular phylogenetic results. The feature of pore occlusion is essential for positioning of new taxa in the cymbelloid clade and for separation of genera with tectula in the broad sense. The importance of pore morphology is supported by molecular data which shows the genera Witkowskia gen. nov., Paraplaconeis and Geissleria monophyletic.

Conclusions

Our results highlight the variation in pore occlusions and their structure in the Cymbellales that has previously been assumed to be a single structure (tectulum). Six new structures are described herein and, with two other, previously described features (annulus and tectulum), these features can be used to diagnose genera within the Cymbellales. Based on the distribution of these features, we propose two new genera to help accommodate the variation in areolar pore occlusion structure. Congruence between the phylogenetic relationships within this order, and the morphological feature of the areolar occlusions within the genera of this group, offers a powerful combination of approaches to understand the relationships within this highly diverse and widely distributed group of freshwater diatoms. These results also offer a more fine-grained taxonomy, based on monophyletic groups, that offer new insights into the diversity and distribution, over both time and space, of this important lineage.

Additional Information and Declarations

Competing Interests

Author Contributions

Data Availability

New Species Registration

The authors declare that they have no competing interests.

Andrei Mironov conceived and designed the experiments, analyzed the data, prepared figures and/or tables, authored or reviewed drafts of the article, and approved the final draft.

Anton Glushchenko performed the experiments, analyzed the data, prepared figures and/or tables, and approved the final draft.

Yevhen Maltsev conceived and designed the experiments, performed the experiments, prepared figures and/or tables, and approved the final draft.

Sergey Genkal performed the experiments, authored or reviewed drafts of the article, and approved the final draft.

Irina Kuznetsova performed the experiments, prepared figures and/or tables, and approved the final draft.

John Patrick Kociolek analyzed the data, authored or reviewed drafts of the article, and approved the final draft.

Yan Liu conceived and designed the experiments, analyzed the data, authored or reviewed drafts of the article, and approved the final draft.

Maxim Kulikovskiy conceived and designed the experiments, analyzed the data, authored or reviewed drafts of the article, and approved the final draft.

The following information was supplied regarding data availability:

We are not providing measurements for each diatom cell, as it is not informative. Measurements for diatom taxa are presented in the manuscript according to a common algorithm, widely accepted in diatomology.

Also, information about amplificated DNA is not included in the manuscript due to being uninformative.

The following information was supplied regarding the registration of a newly described species:

Placoneis elinae sp. nov., Witkowskia gen. nov., Chudaevia gen. nov., Chudaevia densistriata sp. nov.

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
