# Peer review of "Reassessment of pore occlusion in some diatom taxa with re-evaluation of Placoneis Mereschkowsky (Bacillariophyceae: Cymbellales) and description of two new genera"

_PeerJ, doi:10.7717/peerj.17278_

## Round 0.1 · original submission · Major Revisions

Dear Dr. Mironov and colleagues:

Thanks for submitting your manuscript to PeerJ. I have now received two independent reviews of your work, and as you will see, the reviewers raised some concerns about the manuscript. However, these reviewers are optimistic about your work and the potential impact it will have on research studying diatom systematics and biology. Thus, I encourage you to revise your manuscript, accordingly, considering all of the concerns raised by both reviewers.

In your revision, please provide the missing information noted by the reviewers, and also elaborate on the issues raised by reviewer 1.

Please note that reviewer 2 has included a marked-up version of your manuscript.

I look forward to seeing your revision, and thanks again for submitting your work to PeerJ.

Best,

-joe

Reviewer 1 ·

Basic reporting

no comment

Experimental design

no comment

Validity of the findings

Manuscript submitted for review “Reassessment of pore occlusion in some diatom taxa with revalution of Placoneis Mereschkowsky (Bacillaryophyceae: Cymbellales) and description of two new genera” prepared by Mironov A. and co-authors focuses on two main points:
1. Description of a new species, which, due to its unique morphological features and with support based on molecular analyses , was classified into the newly created genus Chudevia.
2. observations of occlusion in representatives of the genus Placoneis and related representatives of the Cymbellaceae family - and, based on them, the establishement of another new genus Witkowskia.


Regarding the first point, I fully agree with the necessity of describing both the species and the genus as new. Both diagnoses are made precisely, they fully document the characteristic features and the documentation is properly illustrated.

However, the second point raises a lot of doubts in me. In my personal opinion, the authors went a step too far in their idea of describing new structures and new genera.
The decision to create a new genus of Witkowski seems to be unjustified. It is based only on the observation of occlusion in various representatives of the Cymbellaceae family. There is no additional support from molecular studies for this idea.

there are no sequences for species that in the opinion of authors should remain in the genus Placoneis (gastrum, amphiboliformis, elinae, coloradoensis). all species for which the sequences were used were considered representatives of the new genus Witkowskia. The only support for the theory comes from the description of occlusions, which are classified into several groups for the purposes of this work. Unfortunately, in my opinion, this classification is far-fetched, and the differences indicated are simply "in the eye of the describer". Comparing the types of occlusion on the plate attached to the manuscript (Fig. 10 A-J), it is difficult to agree that the structures described as tectulum (Fig. 10a), parvutectulum (Fig.10d) and paratectulum (Fig. 10g) are different enough to be treated as separate. In my opinion, the minimal differences between them may result from the fact that they were simply observed in different species, which does not mean that they must constitute separate genera. Supporting the idea of separating several different genera just on the basis of occlusion morphology should be supported the observation of these structures in many different species representing the proposed genera.
For me there is lack of justification for establishing is of a new genus (Witkowskia). Until getting better evidence, this proposal for taxonomic changes for almost all members of the genus Placoneis is unfounded.
Additionally, some elements of the work are prepared chaotically. It is difficult to know which sample and slide correspond to which materials used in the research, i.e.
Figure 2. A3D. Placoneis elinae sp. nov. (Fresh) Sample no. B445 (corresponds material in the slide no. 13899). Light microscopy. Live cells with chloroplast structure.
I do not find those numbers in list of collected samples (tab. 1).
the authors should also verify the prepared work, in several places I noticed the missing year of the cited literature.

I believe that the submitted manuscript can be published in PeerJ, but it requires significant changes and corrections.

·

Basic reporting

no comment

Experimental design

no comment

Validity of the findings

no comment

Additional comments

The MS is about Placoneis sensu lato. The authors tried to solve some taxonomical problems in the group. They proposed two additional genera by using detailed molecular and morphological features of the forms. Images are very good and representable to understand the details. Also, they combined many taxa in a new genus. Please kindly find some remarks in the PDF file.

---

## Round 0.2 · accepted · Accept

Dear Dr. Mironov and colleagues:

Thanks for revising your manuscript based on the concerns raised by the reviewer. I now believe that your manuscript is suitable for publication. Congratulations! I look forward to seeing this work in print, and I anticipate it being an important resource for groups studying diatom systematics and biology. Thanks again for choosing PeerJ to publish such important work.

Best,

-joe

Reviewer 1 ·

Basic reporting

no comment

Experimental design

no comment

Validity of the findings

no comment

Additional comments

the authors made corrections to the article indicated in the previous review. they also included a detailed explanation of the previous comments.
The article can be published in its current form